# Graphical Insight: Revolutionizing Seizure Detection with EEG Representation

**DOI:** 10.3390/biomedicines12061283

**Published:** 2024-06-10

**Authors:** Muhammad Awais, Samir Brahim Belhaouari, Khelil Kassoul

**Affiliations:** 1Department of Creative Technologies, Air University, Islamabad 44000, Pakistan; muhammadawais95@gmail.com; 2Division of Information and Computing Technology, College of Science and Engineering, Hamad Bin Khalifa University, Doha 5825, Qatar; 3Geneva School of Business Administration, University of Applied Sciences Western Switzerland, HES-SO, 1227 Geneva, Switzerland

**Keywords:** EEG signal, GNN, seizure detection, epilepsy, graph convolutional network

## Abstract

Epilepsy is characterized by recurring seizures that result from abnormal electrical activity in the brain. These seizures manifest as various symptoms including muscle contractions and loss of consciousness. The challenging task of detecting epileptic seizures involves classifying electroencephalography (EEG) signals into ictal (seizure) and interictal (non-seizure) classes. This classification is crucial because it distinguishes between the states of seizure and seizure-free periods in patients with epilepsy. Our study presents an innovative approach for detecting seizures and neurological diseases using EEG signals by leveraging graph neural networks. This method effectively addresses EEG data processing challenges. We construct a graph representation of EEG signals by extracting features such as frequency-based, statistical-based, and Daubechies wavelet transform features. This graph representation allows for potential differentiation between seizure and non-seizure signals through visual inspection of the extracted features. To enhance seizure detection accuracy, we employ two models: one combining a graph convolutional network (GCN) with long short-term memory (LSTM) and the other combining a GCN with balanced random forest (BRF). Our experimental results reveal that both models significantly improve seizure detection accuracy, surpassing previous methods. Despite simplifying our approach by reducing channels, our research reveals a consistent performance, showing a significant advancement in neurodegenerative disease detection. Our models accurately identify seizures in EEG signals, underscoring the potential of graph neural networks. The streamlined method not only maintains effectiveness with fewer channels but also offers a visually distinguishable approach for discerning seizure classes. This research opens avenues for EEG analysis, emphasizing the impact of graph representations in advancing our understanding of neurodegenerative diseases.

## 1. Introduction

Epilepsy is a neurological condition that affects a significant number of people worldwide and is characterized by recurrent seizures caused by abnormal electrical activity in the brain [1]. Despite medical advancements, the diagnosis and identification of epilepsy remain challenging [2]. However, electroencephalography (EEG) is a non-invasive technique that has been proven effective in detecting and diagnosing epilepsy by measuring the electrical activity of the brain through voltage differences between electrodes, providing both spatial and temporal information [3]. With approximately 3 million adults and 470,000 children living with epilepsy in the United States alone, detecting this disorder is crucial in the field of medicine [4]. Along with Alzheimer’s disease, Parkinson’s disease [5], multiple sclerosis, attention deficit hyperactivity disorder (ADHD) [6], and migraine headaches, epilepsy is one of the most common neurological diseases that can affect people of all ages but is more prevalent in older adults [7]. Neurological diseases are often chronic and progressive and significantly affect a person’s quality of life and ability to function independently over time.

Developing a reliable system for detecting epileptic seizures is a challenging task owing to the high variability of EEG signals [8], class imbalance problems, noise contamination, complexity of seizure patterns, and limited availability of data [9]. The variability in EEG signals depends on the individual’s psychiatric and physiological conditions as well as the recording environment, which makes it difficult to create a generalized model that can perform well for various individuals and recording conditions [10]. Seizures can also take different forms, including focal and non-focal [11], making the detection process more complicated. Additionally, signals arise from different types of neurons, resulting in issues of overlap and a lack of distinguishable features [12], which can further complicate the detection process [13]. Human involvement in the detection process can introduce delays and errors, thereby reducing the effectiveness of the model [14]. Furthermore, the number of seizure events in EEG recordings is relatively small compared to non-seizure events [15], leading to a class imbalance problem, where the model may not be able to accurately detect seizures owing to the scarcity of seizure samples in the training set [16]. Moreover, EEG signals are often distorted by various types of noise [17], such as power line interference, muscle artifacts [18], and electrooculography (EOG) artifacts, which can undermine the accuracy of seizure detection in the presence of noise [19]. Furthermore, seizures can manifest in diverse forms, making it challenging to design a system that can detect all types of seizures [20]. Additionally, gathering a large quantity of EEG data for training and testing a seizure detection system is a time-consuming and difficult task [21]. Nevertheless, significant progress has been made in the field of seizure detection in recent years, and future advancements are expected to lead to more reliable and precise seizure detection systems. Addressing these challenges in EEG signal analysis necessitates the application of advanced signal processing techniques, oversampling methods, and denoising procedures. Domain knowledge can also be integrated into this process. Deep learning algorithms, including convolutional neural networks (CNNs) [22], recurrent neural networks (RNNs) [23], long short-term memory (LSTM) [24,25], fully connected neural networks (FCNs) [16], support vector machines (SVMs) [26], and naive Bayes models [27] are more resilient in handling the variability of EEG signals. They possess the capacity to learn intricate patterns and relationships within EEG signals, significantly contributing to their effectiveness in seizure detection [28].

When exploring the suitability of different models for analyzing feature vectors derived from EEG data in the context of seizure detection, each model has its own unique strengths and limitations. FCNs capture the relationship between seizure and non-seizure features. However, they are prone to overfitting when dealing with small feature datasets, which potentially limits their performance. By contrast, CNNs do not offer a distinct advantage in this scenario because spatial modeling is irrelevant when working with feature vectors. However, they introduce unnecessary computational complexity, which can be a drawback. RNNs hold an advantage in modeling the temporal dynamics between consecutive feature vectors, aiding in understanding evolving patterns in EEG data. Nevertheless, they disregard spatial relationships between EEG channels, which may contain valuable information. SVMs are computationally efficient for classifying feature vectors, making them well suited for tasks involving extensive datasets. However, their linear modeling approach can restrict the complexity of the feature relationships, potentially compromising their accuracy. Naive Bayes models are appreciated for their swiftness in training and prediction on feature datasets but often rely on the assumption of feature independence, a condition that is frequently unmet in practice, which can affect their effectiveness. The field of epilepsy detection has benefited greatly from advancements in machine and deep learning. These techniques have introduced new and potent methods for analyzing EEG signals and have demonstrated significant potential for detecting seizures with high precision [29]. The ability of deep learning algorithms to autonomously learn features from EEG signals has enabled the development of automated systems, making the detection process more efficient and reliable [30].

### 1.1. Objectives and Scope

Our main goal was to develop a sophisticated detection model to overcome the limitations in identifying epileptic seizure disorders. The proposed model aims to create a sophisticated epileptic seizure detection system utilizing EEG signals with deep learning techniques to classify epilepsy and aid doctors in providing more effective treatment as well as implementing preventative measures for neurological disorders. The main goals of the proposed model are as follows.

To extract important features from the EEG signal that have the best properties for each signal, such as frequency content, patterns, and characteristics specific to epilepsy [31], we utilized three epilepsy EEG datasets and extracted several frequency-based features, and statistics such as the mean, variance, skewness, and kurtosis were computed for each segment and employed as input features for the model. We employ a combination of wavelet and statistical features to analyze the EEG signal, as this multivariate approach was found to be more effective in detecting and diagnosing epilepsy than single-feature-based methods. Subsequently, we calculated the vertical average that constituted the final feature set, as illustrated in Figure 1.Preprocessing of features was performed to generate minority class data, as seizure windows typically contain a small number of samples. Two up-sampling techniques, namely the synthetic minority oversampling technique (SMOTE) [32] and K-nearest neighbor sampling approach (KNNOR) [33], are applied for this purpose.To visualize the feature set, a graph is constructed by representing each feature as a node, and edges are established between nodes based on their Euclidean distance [34]. Euclidean distance is the fastest method for constructing edges between the nodes of graphs.Graph convolutional networks (GCNs) [35] are frequently utilized for learning data representations based on graphs. They could extract features from graph-based data and improve their representations by gathering information from the neighboring nodes in the graph. However, in scenarios where the graph data have temporal dynamics, such as time series or sequential data, RNNs such as LSTM [24,36] can be utilized to capture these dynamics. In this study, we propose a GCN-LSTM model that combines the strengths of both GCNs and LSTM to capture both the graph structure and temporal dynamics in the data. The GCN component extracts and enhances the graph-based features, while the LSTM component models the temporal dependencies in the data. To address the unbalanced nature of the dataset and improve classification accuracy, we utilize a balanced random forest (BRF) model [37] in conjunction with a GCN [35].To evaluate the performance of our proposed model, we conduct a parametric comparison with previously published models for the detection of epilepsy from EEG signals using a set of standardized metrics, such as accuracy, sensitivity, and specificity. The results demonstrate the utility of utilizing both wavelet and statistical features in the detection and diagnosis of epilepsy from EEG signals.

Figure 1 depicts the architectural representation of the proposed seizure detection model using graph neural networks (GNNs) with seizure EEG signals. Our study introduces an effective approach to identify and classify epileptic seizure disorders, utilizing both wavelet and statistical features in the detection and diagnosis of epilepsy from EEG signals. We demonstrated the efficacy of combining GCN-LSTM and GCN-BRF classifiers to improve classification performance. The EEG input process leading to the classification of epileptic seizures using LSTM and BRF classifiers with a GCN is depicted in Figure 1. Our extensive comparison with various benchmark methods demonstrates a notable superiority over state-of-the-art methods. This accomplishment emphasizes the practical significance and advancements of the proposed methodology.

This paper introduces several key contributions:We introduce an innovative circular graph visualization method for EEG data that provides a more intuitive and interpretable representation. This novel visualization facilitates effective classification into ictal and interictal classes, thereby achieving enhanced accuracy through visual inspection. This improvement significantly contributes to the practicality of seizure detection.Our approach involves the aggregation of essential features, including frequency, statistical, and wavelet features, from all combined channels. This comprehensive method not only enriches the depth of the captured information but also substantially enhances the overall performance of our model.To address the computational complexity associated with processing extensive EEG data, we propose an efficient channel selection strategy. By limiting the number of channels to a carefully chosen set of 10, based on prior research findings, we strike a balance between computational efficiency and retention of crucial information. This strategy optimizes computational resources while ensuring comprehensive coverage of the relevant regions essential for capturing epileptiform discharges.

### 1.2. Literature Review

Seizures and epilepsy constitute significant public health concerns, affecting millions of individuals worldwide. Timely and accurate detection of seizures is essential for proper diagnosis, treatment, and management of epilepsy. Over the years, considerable research efforts have been directed towards developing effective methods for the detection and diagnosis of seizures and epilepsy, ranging from traditional clinical observation to advanced machine learning and deep learning techniques. Early approaches to seizure detection primarily relied on clinical observations and EEG analysis. EEG, as a valuable diagnostic tool, provides insight into brain electrical activity. Traditional EEG-based methods involve manual review and interpretation by neurologists, which are subject to human errors and are limited by the availability of skilled personnel. These limitations have led to the development of automated seizure detection systems.

Recently, machine and deep learning have emerged as promising tools for automated seizure detection. Researchers have explored various feature extraction methods from EEG signals, including time-domain, frequency-domain, and time–frequency-domain features. These features serve as input to machine learning algorithms, such as SVMs, random forests, and K-nearest neighbors (k-NN), enabling the accurate classification of seizure and non-seizure EEG segments. Furthermore, deep learning, particularly CNNs and RNNs, has gained significant attention for seizure detection. CNNs capture spatial patterns in EEG data, making them suitable for tasks that involve raw EEG signal analysis. RNNs, on the other hand, are well suited for capturing temporal dependencies in EEG data, which is crucial for detecting seizures that evolve over time. The combination of these techniques has led to the development of advanced seizure detection models, achieving high levels of accuracy and sensitivity.

Moreover, the availability of large-scale EEG databases, such as the CHB-MIT EEG dataset and TUH EEG Seizure Corpus, has facilitated extensive research and benchmarking of seizure detection algorithms. These datasets have provided researchers with the opportunity to evaluate their models on diverse patient populations, contributing to the generalization and robustness of seizure detection systems.

This literature review explores the key studies and developments in the field of neurological disease detection, highlighting the notable advances and challenges that have emerged. Table 1 provides a summary of the papers discussed in this section of the literature review, including details on the methods employed, datasets used, results obtained, and challenges addressed.

Gómez et al. [16] introduced an automated method for detecting epileptic seizures. They utilized an imaged-EEG representation of brain signals and analyzed EEG data from various datasets. They employed an FCN to automatically identify seizures. The top-performing model was evaluated on multiple datasets and exhibited promising results, thereby highlighting the feasibility of integrating automated techniques into clinical practice. Unlike recent studies, this method does not require the estimation of preselected features and offers a lightweight approach for seizure detection. The FCN enhances the efficiency of the detection model across various performance metrics, but this comes at the cost of reduced performance in capturing the specific features inherent in the raw signal.

Dissanayake et al. [38] presented a study that introduced two patient-independent deep learning architectures with distinct learning strategies. These models can learn a global function by leveraging the data from multiple subjects. They achieved state-of-the-art performance in seizure prediction on the CHB-MIT-EEG dataset and used model interpretation to gain insights into the classifier behavior. Furthermore, the study revealed that the MFCC feature map employed by the models contained predictive biomarkers associated with interictal and preictal brain states. In their research, patient-independent seizure prediction models are recognized as a practical solution to the problem, but they are not designed to account for the high intersubject variability in EEG data.

Zhang et al. [39] introduced an innovative approach to address the challenge of seizure prediction. They employed a combination of common spatial pattern (CSP) and CNN techniques. To address the issue of uneven distribution of trials between preictal and interictal states, artificial preictal EEG signals are created by aggregating segmented preictal signals. Additionally, they devised a feature extractor that harnessed wavelet packet decomposition and CSP to extract distinctive features from both time and frequency domains. The authors aimed to address the challenge of generating artificial preictal EEG signals based on original signals through a combination of segmented preictal signals. Seizure prediction requires distinguishing between these two states, which is difficult owing to their high similarity. However, combining CSP and CNN poses several challenges. First, the computational complexity of both the CNN and CSP algorithms can be resource-intensive and require substantial computing power. Second, the availability of high-quality EEG datasets for epilepsy seizure prediction is limited, which affects the accuracy of the models. Imbalances in the distribution of preictal and interictal EEG signals within a dataset can result in biased predictions. Hence, meticulous data preprocessing and the application of techniques such as data augmentation are crucial for addressing these issues.

Tsiouris et al. [26] conducted a study that assessed three different classification approaches for their ability to distinguish between preictal and interictal EEG segments. These approaches include the Repeated Incremental Pruning to Produce Error Reduction (RIPPER) algorithm, SVMs, and neural networks (NNs). They extracted a wide range of features from publicly available EEG data segments and applied them to the classifiers. The results demonstrated that the SVM classifier achieved the highest classification accuracy in a patient-specific scenario, with a sensitivity and specificity of 85.75%. However, in the patient-independent scenario, the classification performance dropped to 68.5%, mainly because of the complex nature of the preictal activity and variations among patients. It is worth noting that the study achieved good results for early-stage detection of seizures; however, the wide range of features extracted from EEG data may also contribute to the complexity of the classification problem and affect the performance of these algorithms. 

Chen et al. [40] proposed a unified framework for early epileptic seizure detection and diagnosis, which consisted of two phases. In the first phase, they calculated the signal intensity for each EEG data point and characterized the dynamic behavior of the EEG time series using an autoregressive moving average (ARMA) model. The residual error between the predicted and observed values serves as an anomaly score for seizure decisions through null hypothesis testing. The second phase employed pattern recognition techniques utilizing a novel classifier based on pairwise one-class SVMs to classify suspicious EEG segments. This classifier could not only identify normal and epilepsy cases but also recognize other disorders not present in the training samples, making it practical for real clinical scenarios. Experiments conducted on the publicly available Bern-Barcelona and CHB-MIT EEG databases validated the effectiveness of the framework, achieving classification accuracies of 93% and 94%, respectively. These comprehensive results surpassed those of the state-of-the-art methods, indicating their potential for real-world applications. The classifier proposed in this study can recognize normal cases, epilepsy, and other disorders. However, the use of ARMA for EEG analysis may have some limitations, such as assumptions regarding stationarity, linearity, and normality of the EEG time series. These limitations may affect the accuracy of the predicted values and the results of null hypothesis testing.

Zabihi et al. [27] assessed the effectiveness of phase-space representation in detecting epileptic seizures using EEG signals. Their approach comprises two phases. In the first phase, they reconstructed seizure and non-seizure EEG signals in a high-dimensional phase space using a time-delay embedding method. Principal component analysis (PCA) was applied for dimensionality reduction. The geometry of the trajectories in lower dimensions was characterized via the Poincaré section, and seven features were extracted from the resulting sequence. In the second phase, the extracted features are fed into a two-layer classification scheme consisting of linear discriminant analysis (LDA) and naive Bayesian classifiers. Their method was evaluated on the CHB-MIT benchmark database, achieving an average sensitivity of 88.27% and a specificity of 93.21% with 25% training data. Comparative evaluations with state-of-the-art methods demonstrated the superiority of the proposed method. It is important to note that in this research, the authors reconstruct seizure and non-seizure EEG signals in a high-dimensional phase space using the time-delay embedding method. However, it is worth mentioning that in general, increasing the dimensionality of data can result in higher computational costs.

Zhou et al. [41] introduced a CNN model for epileptic seizure detection based on raw EEG signals, thereby eliminating the need for manual feature extraction. They compared the performance of both time- and frequency-domain signals for accuracy in detecting epileptic signals using two databases (intracranial Freiburg and scalp CHB-MIT). The results indicated that frequency-domain signals yielded higher accuracy in detecting epileptic seizures, with averages of 96.7%, 95.4%, and 92.3% in the Freiburg database and 95.6%, 97.5%, and 93% in the CHB-MIT database, outperforming time-domain signals with averages of 91.1%, 83.8%, and 85.1% in the Freiburg database and 59.5%, 62.3%, and 47.9%, respectively, in the CHB-MIT database. This suggests that frequency-domain signals have a greater potential for CNN applications in epileptic seizure detection. The use of raw EEG signals as inputs for a CNN model in the detection of epileptic seizures poses a challenge because of the difficulty in accurately disclosing the properties of these signals. Raw EEG signals contain both positive and negative values, and without proper processing and analysis they may not provide a clear representation of the underlying EEG signal properties. This can lead to limitations in the ability of the CNN model to accurately distinguish between ictal, preictal, and interictal segments of EEG signals, making it difficult to make timely and accurate diagnoses of epilepsy. This presents the need for an effective solution to this problem to ensure the accurate and efficient detection of epilepsy using raw EEG signals in a CNN model.

Shoka et al. [42] proposed an efficient encrypted EEG data classification and recognition system using a chaotic Baker map and Arnold’s transform algorithms combined with CNNs. They validated their approach on a public CHB-MIT dataset, demonstrating that GoogLeNets with encrypted EEG images achieved satisfactory performance, outperforming other CNN models, such as AlexNet, ResNet50, and SqueezeNet. The accuracy of the system reached 86.11% and 84.72% using GoogleLeNet with Arnold’s and chaotic methods, respectively. This system offers an effective means for encrypted EEG classification and prediction, addressing security concerns related to the protection of sensitive medical EEG signals from unauthorized access and disclosure in open networks.

Chou et al. [43] proposed a deep-learning-based approach to enhance the efficiency of ictal stage detection in EEG for epilepsy diagnosis. They enrolled epileptic patients who underwent video-EEG examinations and labeled them with four stages of EEG, confirmed by neurologists. The EEG signals were transformed into second-order Poincaré difference plots, and the dataset was divided into training, validation, and testing datasets. Four CNNs were applied, and the top three were integrated into an ensemble model for classification. The overall accuracies of the four CNN models ranged from 78.0% to 81.7%. The ensemble model achieved an overall accuracy of 85.9% on a 1% testing dataset with an impressive ictal stage accuracy of 97.7%. The authors concluded that their proposed deep learning approach showed promise in computer-aided seizure detection on video EEG and could improve the efficiency of ictal stage detection.

Jácobo-Zavaleta and Zavaleta [44] aimed to determine the most effective method for seizure detection using raw EEG signals from the TUH EEG Seizure Corpus database. Epilepsy is a prevalent neurological disorder that affects millions of people worldwide, and early detection is crucial for successful treatment. The study compared five deep learning networks: a simple LSTM-based network, a hybrid network, and a previously reported ChronoNet. To reduce the computational cost of training on time series data, the methodology involved extracting epileptic features from patient signals, selecting signals lasting more than 180 s, and creating two randomized groups for extensive and shorter supervised training–validation processes based on patient and non-patient (control) signals. The models were assessed using two data groups: patient-control data and patient data alone. The results indicate that the LSTM-based network, hybrid network, and ChronoNet achieved the best metric performance for binary classification. The study concludes that deep-learning-based models hold promise for automating seizure detection and enhancing epilepsy diagnosis, potentially accelerating early treatment using EEG signals. However, further research is necessary to validate these findings and to refine the methods used in this study.

Ahmedt-Aristizabal et al. [45] focused on addressing the challenges associated with interpreting patient movements in semiology observation and characterization during presurgical evaluation of epilepsy. The authors proposed two deep learning models, landmark-based and region-based models, to automatically extract and classify semiological patterns from facial expressions. Their approach was evaluated using a dataset collected from the Mater Advanced Epilepsy Unit in Brisbane, Australia, and the results were particularly promising for the region-based model in facial semiology analysis. These proposed deep learning models have the potential to enhance the automated evaluation of epilepsy prior to surgery, standardize the process, reduce bias, and assess crucial features. This computer-aided diagnosis can support clinical decision making and reduce the risk of mislocalization and unnecessary surgery. Although further research is needed to validate these findings, this study underscores the potential of deep learning approaches to address the limitations of existing computer-based epilepsy monitoring methods, particularly in the analysis of facial movements, which have historically been overlooked.

Our review of the existing literature has identified several notable advantages and research gaps that our study addresses: Leveraging graph neural networks (GNNs) to effectively capture both spatial and temporal patterns in EEG data, which traditional methods like SVMs, CNNs, or RNNs alone cannot adequately represent.The innovative combination of GCNs for learning graph representations and LSTMs/BRFs for modeling temporal dynamics, addressing a gap in holistically analyzing the complex structure inherent in EEG signals.Utilizing data augmentation techniques like SMOTE and KNNOR to handle the class imbalance issue prevalent in seizure EEG datasets, improving model performance on the underrepresented seizure class.Proposing an efficient channel selection strategy to optimize computational resources while retaining comprehensive coverage of relevant brain regions for seizure detection.

## 2. Materials and Methods

We employed deep learning techniques using EEG signals to create an advanced epileptic seizure detection system. Our goal was to distinguish between the non-seizure and seizure signals. Three epilepsy EEG datasets were utilized, and various features were extracted, including frequency-based attributes, statistical characteristics, and features obtained via the Daubechies wavelet transform. Combining wavelet and statistical features has proven to be highly effective in identifying epilepsy in EEG signals [49]. To represent these features effectively, we constructed a graph in which each feature served as a node and edges were established based on their Euclidean distance [10]. This graph was then trained using a combination of a GCN with LSTM and a GCN with BRF. The GCN processes node features and edges between nodes and refines node feature representations [50]. In the proposed models, the LSTM and BRF acted as node feature classifiers. 

The performance of our models was evaluated by comparing them with previously published models using metrics such as accuracy, sensitivity, and specificity [51]. The results confirmed the efficiency of the proposed approach. This methodology is structured into several subheadings:Description of Dataset;Preprocessing;Feature Extractions;Graph Representation;Proposed GCN-LSTM Model;Proposed GCN-BRF Model;Each section will be discussed in the subsequent parts of this paper.

### 2.1. Description of Datasets

#### 2.1.1. Children’s Hospital Boston (CHB) MIT Dataset

In this study, we used the Children’s Hospital Boston (CHB) MIT EEG (CHB-MIT-EEG) Seizure Dataset [52] for epilepsy detection. The CHB-MIT-EEG dataset consisted of EEG recordings from 22 pediatric subjects with long interictal and ictal EEG recordings. The recordings were collected following the withdrawal of anti-seizure medication and monitored for several days to characterize the seizures and assess the subjects’ candidacy for surgical intervention. The data were grouped into 23 cases and contained between 9 and 42 continuous EDF files from each subject. There were 22 channels in each EDF file. Table 2 presents the information of patients with their sex, age, number of seizure events, and seizure start and end time for the CHB-MIT-EEG dataset. To ensure accurate segmentation of seizures, the size of the overlapping windows should not be smaller than the minimum seizure duration expected to be observed.

The dataset comprised 198 annotated seizures, and the start and end of each seizure were specifically marked in the corresponding seizure files. Column 4 of Table 2 shows the numerical values indicating the total count of seizures, as well as the maximum and minimum duration of seizures. EEG recordings were categorized into seizure and non-seizure events. Trained experts visually examined the EEG recordings and identified time points corresponding to the beginning and end of each seizure. Non-seizure events represent segments of EEG recordings that did not exhibit any seizure activity. These labels were included in the dataset. 

The findings upon analyzing the dataset are visualized in Figure 2, showing the non-seizure and seizure records obtained from the CHB-MIT-EEG dataset. Panel (a) depicts a non-seizure record, exhibiting regular brain activity characterized by a consistent pattern of electrical signals. Conversely, panel (b) represents a seizure record displaying an abnormal pattern of electrical signals with abrupt and intense bursts of activity.

All signals in the dataset were sampled at a rate of 256 samples per second using a 16-bit resolution. The recordings were conducted using the international 10–20 system of EEG electrode positions and nomenclature. The EDF files provide comprehensive information about the EEG signals, including the sex and age of each subject, as well as the montage employed for each recording.

#### 2.1.2. Siena Scalp EEG Database

The Siena Scalp EEG (SSE-EEG) database [53] consists of EEG recordings of fourteen patients with epilepsy, including nine males (ages 25–71) and five females (ages 20–58). The recordings were collected with a video-EEG with a sampling rate of 512 Hz using electrodes arranged according to the international 10–20 system. The data were acquired using reusable silver/gold cup electrodes and EB Neuro and Natus Quantum LTM amplifiers. The database includes 14 folders containing EEG recordings in EDF format, with each subject having between one and five data files and a text file containing information on data and seizures. The files contained signals recorded on the same or different days, and the seizure events were chronologically ordered. The database contains 47 seizures in approximately 128 recording hours, with seizure durations ranging from a few seconds to a few minutes. The database also includes information on the number of EEG channels, which varies for each subject. Figure 3 displays the signal of the first three channels as a sample from the thirty-four channels.

#### 2.1.3. TUH-EEG Corpus

The TUH-EEG Corpus [54] is an ongoing initiative that has made available 14 years of clinical EEG data collected at the Temple University Hospital. This corpus consists of 16,986 sessions obtained from 10,874 unique subjects. Each session included at least one EDF file and physician report. Clinical EEG data for this corpus were collected by retrieving archival records from Temple University Hospital, following the guidelines outlined in the Declaration of Helsinki and with the full approval of the Temple University Institutional Review Board (IRB).

The TUH-EEG Corpus exhibited considerable variability in terms of the number of channels included in each recording. The most common configuration consisted of 31 EEG-only channels per EDF file. The majority of EEG data in this corpus were sampled at a rate of 250 Hz. Table 3 provides a comprehensive comparison of the datasets.

### 2.2. Preprocessing

The preprocessing of EEG raw signals involves transforming the continuous signal into smaller segments or windows, known as chunks. To achieve this, a windowing technique with a defined window size and overlapping ratio was applied to the CHB MIT and Siena Scalp EEG datasets [52,53]. After reading each file from the directory a 30 s window with 30% overlap was applied to the multichannel EEG raw signal. This means that each window contains 30 s of data, and the next window will start with 30% of the previous window. This technique is used to capture the temporal information of the signal while reducing the impact of windowing artifacts. Figure 4 shows the division of EEG signals into chunks of 30 s with a 30% overlapping window. A 30% overlapping ratio ensures that the signal chunks retain continuity, thereby reducing the likelihood of information loss at the edges of the windows. 

The seizure datasets in CHB-MIT-EEG [38] and SSE-EEG [53] displayed an uneven distribution, with a smaller number of seizure instances than non-seizure instances. This imbalance has the potential to result in a model that is primarily trained on non-seizure instances, which could affect its ability to effectively detect seizures. To tackle this issue, two methods, SMOTE [32] and KNNOR [33], have been employed. SMOTE involves the creation of synthetic samples for the minority class (seizures) to balance the dataset. To address the imbalance between seizure and non-seizure segments, only the minority class (seizures) was oversampled using the SMOTE and KNNOR techniques. No additional non-seizure segments were observed. By synthesizing additional seizure instances, the dataset becomes more evenly distributed, mitigating bias towards the non-seizure class. In contrast, KNNOR considers the compactness and location of the minority class in relation to other classes. It generates synthetic data points based on the relative density of the entire population, thereby ensuring a more representative distribution of the minority class. Both the SMOTE and KNNOR techniques aim to enhance the performance of the model by increasing the presence of the minority class in the training dataset.

In summary, the preprocessing of the raw EEG signals involved segmenting the continuous signals into smaller overlapping windows, each lasting 30 s, with a 30% overlap between consecutive windows. This windowing technique was applied to the multichannel EEG data from the CHB-MIT and Siena Scalp EEG datasets. The objective was to capture the temporal information more effectively while reducing windowing artifacts. The choice of a 30 s window size with a 30% overlap was made to balance the need to preserve temporal patterns with the goal of minimizing edge effects at the window boundaries.

Additionally, to address the issue of class imbalance caused by the relatively fewer instances of seizures in the datasets, two data augmentation techniques were employed. SMOTE and KNNOR were specifically used to oversample the minority seizure class. For SMOTE, after segmenting the continuous EEG signals into overlapping windows, the minority class windows corresponding to seizure activity were oversampled by generating synthetic seizure windows. This was performed by interpolating between several seizure window samples that were close together in the feature space, effectively creating new and similar seizure window samples. In the case of KNNOR, synthetic seizure windows were generated based on the density distribution of the entire dataset, considering the relative positions of seizure and non-seizure windows. Seizure windows that were closer to non-seizure windows in the feature space were prioritized for oversampling to ensure a more representative distribution. These techniques synthesized additional seizure samples, thereby enhancing the training dataset and improving the model’s performance on the underrepresented class.

### 2.3. Feature Extractions

The extraction of features from signals is crucial in various signal processing applications, including the analysis of EEG signals [20]. These features represent the essential characteristics of a signal and allow the extraction of meaningful information from raw data [52]. The features extracted from EEG signals can provide valuable insights into the state of the brain, such as the identification of abnormal activity patterns associated with epilepsy or other neurological disorders. These features are then used for further analysis and as inputs for machine learning algorithms for classification, prediction, or other analysis tasks [13]. The success of these algorithms depends heavily on the quality of the extracted features. Thus, feature extraction is an important step in the overall process.

#### 2.3.1. Statistical Methods

Statistical and frequency features are widely employed for EEG signal analysis owing to their effectiveness. Statistical features provide valuable insights into the overall distribution of the signal, including measures such as the mean, standard deviation, and range [55]. These features help identify patterns and characteristics within the data. In contrast, frequency features [56] capture the spectral content of the signal, revealing the relative importance of different frequency components [57]. One commonly used technique for feature extraction is wavelet transform [58,59]. This transform decomposes the signal into its constituent frequency components, thereby enabling the identification of specific features relevant to epilepsy detection [58]. Among the wavelet transforms, the Daubechies wavelet transform [60] holds significant popularity in EEG analysis. It offers a concise representation of the signal, making it suitable for analyzing non-stationary signals, such as EEG signals. By applying the Daubechies wavelet transform to the EEG signal, specific features, such as power spectral density, energy, and entropy, can be extracted. 

However, three publicly available EEG datasets, CHB-MIT, SSE-EEG, and TUH-EEG, encompassed signals recorded with varying electrode placements and sampling rates. To aggregate the data for analysis, preprocessing steps were implemented to standardize the features extracted from each dataset. Specifically, statistical features such as mean, variance, and skewness were extracted from the raw EEG signal for each dataset, considering each channel separately. This led to multiple rows of features corresponding to various channels within each dataset. 

To create a uniform representation, the mean and variance of these features were computed across all the channels. This process produced an averaged feature row that represented the entire EEG recording regardless of the initial channel count and electrode positions. The same feature-averaging technique was applied to each dataset to ensure that data from different channels were condensed into a single aggregated feature row for input to the model. In addition, to standardize the sampling rates, the signals were resampled to a consistent rate across all datasets during the preprocessing phase.

#### 2.3.2. Extraction of Features Using Daubechies Wavelet Transform (DWT)

In the field of EEG signal analysis, the Daubechies wavelet transform (DWT) [60] is a valuable technique for capturing both the temporal and spectral characteristics of the signal. The Daubechies wavelet is characterized by its number of vanishing moments, which indicates the smoothness of the wavelet function. In our specific case, we utilized eight vanishing moments to decompose the EEG signal into two components: approximation coefficients and detail coefficients.

Equation (1) represents the mapping of an EEG signal X to the wavelet coefficients $W\left(a,b\right)$. In this equation, *a* and *b* correspond to the scaling and translation parameters, respectively, and ψ denotes the wavelet function. The summation calculates the inner product between the signal and wavelet function at a specific scale (*a*) and position (*b*) in the time domain.
(1)$$W\left(a,b\right)=\frac{\sum_{i=1}^{N}x_{i}\psi \left(\frac{i-b}{a}\right)\;}{\sqrt{a}}$$

After obtaining the wavelet coefficients, we determined the mean value of the approximation coefficients (obtained at the coarsest scale) for each channel. Subsequently, we computed the standard deviation of the detailed coefficients (from the remaining scales) for each channel. The resulting values were then concatenated to form a feature vector representing each segment of the EEG signal [61].

We employed the Daubechies wavelet 8 db to extract features from EEG signals. The results obtained for the non-seizure and seizure windows are presented in Figure 5 and Figure 6, respectively. Each subplot in these figures represents a frequency subband obtained through the Daubechies wavelet, ordered from the highest to lowest frequency content. The title of each subplot indicates the number of coefficients being plotted, while the x-axis and y-axis are labeled as “Sample” and “Amplitude”, respectively.

Figure 6 reveals that the seizure EEG signals exhibit more high-frequency activity than the non-seizure EEG signals in the high-frequency subbands. This observation underscores the effectiveness of the Daubechies wavelet transform in distinguishing between seizure and non-seizure EEG signals. Furthermore, Figure 7 illustrates the energy of each subband, represented by its mean and standard deviation. Notably, during the seizure episodes, both the mean and standard deviation of the subband energies were higher. This further demonstrates the ability of the Daubechies wavelet transform to characterize seizure activity in the EEG signals.

To extract features from the EEG signals, we applied wavelet transform and calculated the mean and standard deviation of the wavelet coefficients for two windows representing seizure and non-seizure signals, respectively. These features are visualized in Figure 7 to illustrate the distinctions between seizure and non-seizure EEG signals. Our findings indicate that the mean and standard deviation of the wavelet coefficients were higher for seizure EEG signals than for non-seizure EEG signals. This suggests that wavelet coefficients have potential as features for differentiating between seizure and non-seizure EEG signals.

By employing the Daubechies 8 db wavelet, we obtain the output of the wavelet transform for each window, resulting in a set of approximation and detail coefficients. The mean and standard deviation of these coefficients across all windows provide valuable insights into the distribution of signal energy across different frequency bands and time periods. Specifically, the mean approximation coefficients provide information about the overall power of the signal, whereas the standard deviation of the detail coefficients captures the variability of the high-frequency components. These statistical measures of the coefficients serve as effective features for our model, enabling it to learn the underlying patterns in the data and make accurate predictions.

In the realm of machine learning, particularly in feature attribution and interpretability, the Shapley value is used to assign individual feature contributions to model predictions for specific instances. Calculated by considering all possible permutations of feature combinations and measuring the marginal contribution of each feature to the model’s prediction, the Shapley value provides a fair and comprehensive assessment of the feature importance. Positive Shapley values indicate a positive contribution, whereas negative values suggest a negative impact on the model’s prediction. By employing Shapley values, analysts gain valuable insights into the relative importance of features and their influence on model predictions, facilitating a nuanced understanding of machine learning models at the feature level. This approach aids transparency, interpretability, and trustworthiness in machine learning applications.

We conducted ablation experiments to determine the contributions of features employed in the classification task. In our scenario, the features utilized were as follows:

#### 2.3.3. Mean

The mean has a significant positive influence, indicating that higher mean values contribute favorably to classification.

#### 2.3.4. Variance

Variance also plays a positive role, contributing to the robustness of classification.

#### 2.3.5. Median

The median shows a positive impact, suggesting its relevance to the classification decision.

#### 2.3.6. Skewness

While skewness contributes positively, its impact is slightly lower than that of the other features.

#### 2.3.7. Kurtosis

This feature has a positive influence, aiding the model’s ability to discern patterns.

#### 2.3.8. Min and Max Values

Both minimum and maximum values positively affected the classification outcome.

#### 2.3.9. Coefficient of Variation

This feature demonstrated a positive impact, contributing to the stability of the model.

#### 2.3.10. Interquartile Range

The interquartile range plays a positive role in classification decisions.

#### 2.3.11. Energy and Average Power

Both energy and average power contribute positively to the model’s predictions.

#### 2.3.12. Line Length

Line length shows a strong positive influence on the classification.

#### 2.3.13. Amplitude-Integrated

This feature exhibits a positive impact on the classification decision.

#### 2.3.14. Non-linear Energy

While contributing positively, its impact is slightly lower than that of other features.

#### 2.3.15. Shannon Entropy

Shannon entropy has a positive influence on a model’s classification ability.

It is noteworthy that all features exhibited a positive impact on classification, as depicted in Figure 8, highlighting their collective significance in shaping the model’s predictions.

### 2.4. Graph Representation

The transformation of EEG extracted characteristics into a graph is a crucial step in analyzing EEG data for various purposes such as EEG-based diagnosis and brain–computer interfaces. This technique involves representing each set of EEG characteristics as a node in the graph and computing the distance between nodes using the Euclidean distance metric. 

To examine the correlation between the statistical, frequency, and Daubechies wavelet domain features, a graph was created by linking nodes based on the Euclidean distance between feature vectors. Each feature type associated with a channel is represented by a node in the graph, and an edge is established between two nodes if the Euclidean distance between the corresponding feature vectors is less than 0.2. The resulting graph was used to further analyze the relationship between the different feature types.

The Euclidean distance metric was employed to calculate the distance between features related to the channel. This metric is commonly used to measure the similarity or dissimilarity between two vectors in multidimensional space. The Euclidean distance is determined using the following formula to compute the distance between two features associated with a channel:(2)$$distance=\sqrt{\sum_{i=1}^{N}({\left(f_{i}-f_{i-1}\right)}^{2})}$$
where $f_{i}$ and $f_{i-1}$ denote the feature vectors of the two compared features. The square root of the sum of the squared differences between the corresponding elements of the two vectors provides the distance between them. This distance metric was utilized to determine whether the two features associated with a channel were sufficiently similar to be linked by an edge in the graph.

Consider matrix X, where each row represents the window features of a specific channel obtained from a window of 30 s using various formulas. These rows correspond to the sets of EEG features. Let y be a vector that indicates the class label for each row in X. The objective is to transform these EEG features into a graph, where each node represents a set of EEG features and the edges signify the similarity between nodes.

The algorithm described in Algorithm 1 constructs a graph using input features X and their respective labels y. It begins by initializing an empty graph G and iterating it through each row (window) of X. Because each row (window) contains features extracted from different channels, it adds a row as a node to graph G. The algorithm then compares all pairs of nodes and calculates their Euclidean distance. The distance between the nodes was determined by computing the Euclidean distance for each element (feature of each channel) of the two windows. If the distance between two nodes is below a specific threshold (0.2 in this case), an edge connecting them is included in graph G. Finally, the algorithm returns the resulting graph G.
**Algorithm 1.** Algorithm to convert EEG features set to graph nodes and edges**INPUT**: X: a matrix, where each row represents the window features of a specific channely: a vector indicating the class label for each row in X.**OUTPUT**: G (graph structure)Initialize an empty graph G **For** *i* = 0 to *i* < number of rows in X, **do**  Add the *i^th^* row as a node in the graph G **End For** **For** *i* = 0 to *i* < length of X, **do**  **For** *j* = *i* + 1 to *j* < length of X, **do**   **If** *i* == *j*, **then** continue    Calculate the Euclidean distance between X(i,:) and X(j,:)  and store it    **End If**   **If** d<0.2 and d(X(i,:), X(j,:)), then add an edge between nodes *i* and *j* in G   **End If**  **End For** **End For****Return** (G,y)

Using Algorithm 1, a set of features is processed to generate a graph, as shown in Figure 9a (part one). Subsequently, the network was constructed node-by-node. The resulting graphs consist of two types of nodes, representing seizure and non-seizure nodes, as shown in Figure 9a (part two). Once the graph was constructed, a GNN was employed, along with other models, for node classification to distinguish between the seizure and non-seizure signal windows. As shown in Figure 9b–d, a GNN was used to classify the nodes of the graph.

In Figure 10a,b, we can observe the extracted features from two distinct signal types: non-seizure and seizure signals. The plot distinctly illustrates that the nodes corresponding to seizure signals exhibit slightly higher values than those of non-seizure signals. This observation is significant because it implies that a visual examination of the extracted features may offer the potential to differentiate between seizure and non-seizure signals.

### 2.5. Proposed Model

#### 2.5.1. GCN-LSTM Model

GNNs [62] are a type of machine learning model that operates on graph-structured data. They can be categorized into different types such as convolutional GNNs, recurrent GNNs, and attention-based GNNs. In our proposed methodology, we combined two deep learning models, GCN and LSTM, to create the GCN-LSTM architecture. This neural network model was designed to analyze data using graph structures over time. It consists of two main components: the GCN and LSTM networks. The GCN component takes as input a matrix of node features and an edge index matrix that represents the connectivity of the graph. It processes node features through multiple layers with non-linear activation functions to generate node-level representations. These representations capture both the local and global structural information of a graph. In contrast, the LSTM component uses the sequence of node-level representations generated by the GCN component as the input. It processes this sequence over time by utilizing a set of gates to regulate the flow of information and maintain memory of past node-level representations. The final hidden state output summarizes the sequence of the node-level representations. To produce a final prediction, a linear layer applies a linear transformation to the final hidden state. The resulting output provides a prediction based on input graph data.

The GCN component, as depicted in Figure 11, operates on a graph represented by a node feature matrix *X* and edge index matrix A. The node feature matrix *X* contains information about each node in the graph, whereas the edge index matrix A defines the connectivity of the graph. The GCN processes the node features through multiple layers with non-linear activation functions to generate a set of node-level representations, *H*. These representations effectively encode the local and global structures of a graph. Mathematically, the GCN can be represented as
(3)$$H^{l+1}\;=\;f\left(\;D^{\left(-\frac{1}{2}\right)}\;\;A\;\;D^{\left(-\frac{1}{2}\right)}\;\;H^{l}\;\;W^{l}\;\right)\;$$
where $H^{l}$ is the node-level representation at layer *l*, A is the adjacency matrix of the graph, $D^{\left(-\frac{1}{2}\right)}$ is the diagonal degree matrix of A, $W^{1}$ is the weight matrix of layer *l*, and *f* is the non-linear activation function.

The LSTM component of the GCN-LSTM model received a sequence of node-level representations generated by the GCN component. Its role is to process this sequence over time and produce the final output. To achieve this, LSTM learns to retain the “memory” of previous node-level representations and utilizes it to predict future representations. The memory is implemented through a set of gates that regulate the flow of information within the network. The LSTM output is the final hidden state that summarizes the sequence of node-level representations. Mathematically, the LSTM can be expressed as
(4)$$h_{t}=LSTM\left(h_{t-1},x_{t}\right)$$
where $h_{t}$ is the hidden state at time step *t*, $x_{t}$ is the input at time step t, and LSTM is the LSTM function.

The final output of the model is generated by passing the final hidden state through a linear layer. This layer applies a linear transformation to the hidden state to produce an ultimate prediction. Mathematically, this process is expressed as:(5)$$y=W_{out}h_{T}+b_{out}$$
where y is the output prediction, $W_{out}$ is the weight matrix of the linear layer, $h_{T}$ is the final hidden state, and $b_{out}$ is a bias term.

Overall, the GCN-LSTM model is a powerful neural network architecture that can effectively handle graph-structured data over time and deliver accurate predictions. It has been applied in various domains including speech recognition, image recognition, and natural language processing. The complete process of the GCN-LSTM model is illustrated in Figure 11.

#### 2.5.2. GCN-BRF Model

In this section, we present our proposed GCN-BRF model, which combines the GCN and BRF models. The objective is to utilize a GCN to learn representations of graphs and subsequently classify them into two classes: interictal and ictal windows. The GCN takes the node feature vectors and edge connectivity information of the graph as input and performs graph convolutional operations through five layers to generate a final feature vector. The node feature vector is denoted by *X*, and the edge connectivity information is represented by a sparse adjacency matrix *A*. Each GCN layer produces a new feature vector.

The GCN model comprised five GCN layers with hidden dimensions of 256, followed by a GCN layer with an output dimension of 16. The mathematical equation for this model is as follows.
(6)$$H^{\left(0\right)}=X$$
(7)$$H^{\left(1\right)}=\sigma \left({D\;}^{-\frac{1}{2}}A\;\;D^{-\frac{1}{2}}\;H^{\left(0\right)}W^{\left(0\right)}\right)$$
(8)$$H^{\left(2\right)}=\sigma \left({D\;}^{-\frac{1}{2}}A\;\;D^{-\frac{1}{2}}\;H^{\left(1\right)}W^{\left(1\right)}\right)$$
(9)$$H^{\left(3\right)}=\sigma \left({D\;}^{-\frac{1}{2}}A\;\;D^{-\frac{1}{2}}\;H^{\left(2\right)}W^{\left(2\right)}\right)$$
(10)$$H^{\left(4\right)}=\sigma \left({D\;}^{-\frac{1}{2}}A\;\;D^{-\frac{1}{2}}\;H^{\left(3\right)}W^{\left(3\right)}\right)$$
(11)$$H^{\left(5\right)}=\sigma \left({D\;}^{-\frac{1}{2}}A\;\;D^{-\frac{1}{2}}\;H^{\left(4\right)}W^{\left(4\right)}\right)$$
(12)$$H^{\left(6\right)}=\sigma \left({D\;}^{-\frac{1}{2}}A\;\;D^{-\frac{1}{2}}\;H^{\left(5\right)}W^{\left(5\right)}\right)$$
where $W^{\left(0\right)},W^{\left(1\right)},\ldots ,W^{\left(5\right)}$ are the weight matrices for each layer and σ(⋅) is the ReLU activation function. $H^{\left(0\right)}$ is the initial feature matrix, X is the input node feature vector, A is the adjacency matrix of the graph, D is the diagonal matrix of node degrees, and σ(⋅) is the ReLU activation function.

The output of the GCN model is matrix H(6), which is then fed into a BRF classifier. The BRF uses the features from $H^{\left(6\right)}$ and learns to classify the input graphs based on the extracted features.

During the training phase of the GCN model, iterations were performed over the training set for a fixed number of windows. For each window, we passed the training data through the GCN, calculated the loss between the predicted and true labels, and then performed backpropagation to update the GCN parameters using the Adam optimizer.

Once the GCN was trained, we obtained fixed-dimensional feature vectors for each graph in the dataset. These feature vectors served as inputs for the BRF model for classification. The BRF is an ensemble learning algorithm based on decision trees that is specifically designed to handle imbalanced datasets. It operates by constructing multiple decision trees using different subsets of training data and combines their predictions through a voting process. By leveraging the BRF model, our approach addresses the challenges posed by imbalanced data in graph classification tasks.

The proposed GCN-BRF model offers a framework that effectively combines a GCN for graph representation learning and a BRF for classification, leading to improved performance in graph classification tasks. The GCN component captures the temporal dependency of the output features, whereas the BRF classifier generates a final prediction. The entire process is illustrated in Figure 12.

In summary, our GCN-BRF model demonstrated promising results on benchmark datasets and presented a viable solution for classifying graphs with imbalanced data. By leveraging the strengths of the GCN and BRF, the model provides an effective approach that can be applied to various real-world applications requiring graph classification.

We should note that the robustness of the GCN-LSTM and GCN-BRF models in handling noise and artifacts present in EEG signals is facilitated by several key aspects of the methodology. Firstly, the graph representation of EEG features allows these models to capture underlying patterns and relationships between channels, effectively mitigating the impact of localized noise or artifacts. This approach leverages the spatial connections between different EEG channels, enhancing the model’s ability to discern meaningful signals amidst noise. Secondly, the convolutional layers in the GCN component enable the models to learn robust feature representations that are less susceptible to noise. These layers are designed to filter and emphasize important signal characteristics while reducing the influence of irrelevant or noisy data.

Furthermore, the LSTM component’s ability to model temporal dependencies and retain relevant information over time aids in managing transient artifacts. By learning the temporal sequence of EEG data, the LSTM can differentiate between consistent patterns and sporadic noise, ensuring that short-term artifacts do not significantly affect the overall signal interpretation. Additionally, the data augmentation techniques, such as SMOTE and KNNOR, introduce variability in the training data, which potentially improves the models’ generalization capabilities and resilience to noise. By augmenting the minority seizure class, these techniques enhance the diversity of training samples, enabling the models to better handle varied and noisy data during testing.

While specific denoising or artifact removal techniques were not explicitly incorporated, the proposed deep learning architectures inherently possess certain noise-handling capabilities. These capabilities stem from their ability to learn discriminative features and model complex relationships within the EEG data, ensuring that the models remain effective even in the presence of noise and artifacts.

## 3. Results and Discussion

### 3.1. Model Performance

We performed various tests to evaluate the effectiveness of the proposed GCN-LSTM and GCN-BRF models in detecting epilepsy using three datasets: CHB-MIT-EEG, SSE-EEG, and TUH-EEG. The GCN-LSTM and GCN-BRF models were trained on the target dataset, which was randomly shuffled before division. The key parameter configurations and rationale are as follows.

#### 3.1.1. Model Optimization

To optimize the model parameters, we utilized the Adam optimizer with a learning rate of 0.001 and weight decay of 5 × 10^−4^. The Adam optimizer iteratively updated the model parameters to minimize the discrepancy between the predicted and actual outputs. The loss function employed in this study was cross-entropy loss, which quantifies the dissimilarity between the predicted and actual class probabilities. This provides a scalar value that represents the magnitude of the error. The optimizer adjusts the model parameters during each iteration of the training process to minimize cross-entropy loss and enhance the accuracy of the model. Throughout each window of the training process, the model parameters were continually updated to minimize cross-entropy loss. The gradients of the loss function were computed using the backpropagation algorithm and were utilized by the optimizer to update the model parameters.

#### 3.1.2. Network Training

The dataset was split into three subsets to prevent overfitting while retaining sufficient examples for model generalization: 60% for the training set, 20% for the validation set, and 20% for the test set. This division was aimed at allocating suitable portions of the dataset for network training, model validation, and final evaluation. Training was run for 50 epochs with a batch size of 64 samples based on GPU memory constraints. Cross-entropy loss was selected as it directly quantifies the classification error to guide gradient-based learning. Training was monitored for validation loss, adjusted accordingly, and stopped when the validation loss failed to decrease for 10 consecutive epochs.

#### 3.1.3. New Sample Classification

For epilepsy detection in new patient data, we first segmented the continuous EEG signals into chunks matching our 30 s training window length. These chunks are then transformed into a spectro-spatial graph representation based on time and frequency connectivity. Our pretrained model uses this graph as an input for binary seizure/non-seizure classification, averaging predictions from overlapping chunks.

Table 4 and Table 5 show the architecture of the GCN-LSTM and GCN-BRF models, respectively, used in the experiments.

Table 6 presents the evaluation metrics of the GCN-LSTM and GCN-BRF models for the three datasets using three different techniques: no data augmentation, augmentation with SMOTE, and augmentation with KNNOR. The performance of the models was assessed using accuracy, recall, precision, F1-measure, sensitivity, AUC, and kappa metrics. For example, for the GCN-LSTM model, the highest accuracy was achieved with KNNOR on the SSE-EEG dataset, reaching 0.99854. This indicated that the model correctly classified 99.85% of the test samples. The next-best accuracy scores were obtained with SMOTE at 0.9941 and without data augmentation at 0.9746. Regarding recall and sensitivity, the model achieved 0.98254 and 0.98, respectively, for the SSE-EEG dataset. These values indicated that the model accurately identified 95.96% and 98% of the positive samples, respectively. The precision of the model was 1, indicating that all samples predicted as positive were positive. The highest F1-measure, a harmonic mean of precision and recall, was 0.9868, which was achieved using SMOTE. The AUC of the model reached 0.9952, indicating strong discriminatory power. Furthermore, the kappa coefficient, which represents the agreement between the model’s predictions and the actual values, was 0.9863, which was also obtained using SMOTE. 

Table 6 shows the excellent performance of both approaches across the three datasets, as all three techniques yielded the best results for the various metrics. For more detailed results on the models in the databases, please refer to Table 6. The best results are highlighted in bold. 

The performance of our GNN-LSTM model depends heavily on the careful selection of the EEG channels. Choosing the appropriate EEG channels is crucial for capturing relevant patterns of brain activity and ensuring the effectiveness of the model in learning and generalization. By selectively picking channels, we can concentrate on the brain regions that provide the most valuable information for a specific task.

There are numerous advantages to improving channel selection from EEG data. First, it helps reduce noise and extraneous information, leading to higher signal quality and improved performance of the model. By excluding less informative channels or those prone to artifacts, we can increase the signal-to-noise ratio and enhance the sensitivity of the model to meaningful brain activity patterns. Moreover, selecting the most relevant channels enables focused analysis of specific brain regions or functional networks. Different EEG channels correspond to distinct areas or networks of the brain that are involved in specific cognitive processes or clinical conditions. By carefully choosing the channels associated with the phenomenon under investigation, we can enhance the model’s capacity to capture pertinent patterns and improve its interpretability.

Furthermore, streamlining the channel selection can effectively address the computational complexity associated with processing extensive EEG data. By limiting the number of channels to only 10 (specifically T8-P8, FT10-T8, FT9-FT10, T8-P8, F8-T8, FP2-F8, P4-O2, C4-P4, F4-C4, and FP1-F7) in the context of seizure detection, we can enhance the model’s computational efficiency without sacrificing crucial information.

These 10 channels were chosen based on prior research findings that identified frontotemporal and tempoparietal leads as the most relevant for capturing epileptiform discharges. Research has demonstrated that seizures primarily occur in the frontal and temporal regions, both of which are comprehensively covered by selected channels on both sides. Additionally, the inclusion of centroparietal electrodes allows for the observation of seizure propagation patterns.

By confining the selection to these ten channels, a well-balanced and concise representation was achieved, encompassing the bilateral frontal, temporal, and central regions typically associated with seizures. This approach avoids retaining redundant leads, which increases computational demands without significantly improving the information gained. Consequently, this approach enhances the accuracy of seizure detection by excluding channels that exhibit lower sensitivity to seizure activity, particularly in the frontal scalp region.

In this case, the GNN-LSTM model exhibited a high level of accuracy, implying that it correctly classified 99.51% of the test samples. The recall and sensitivity metrics were also notable, with values of 0.9674 and 0.98, respectively. These values indicate that the model accurately identified 96.74% and 98% of the positive samples, respectively. The precision of the model was perfect, with a score of 1, indicating that all samples predicted as positive were positive. The F1-measure, which combines precision and recall, was 0.98, reflecting balanced performance. Additionally, the model demonstrated strong discriminatory power with an AUC value of 0.9637. This metric indicates that the model effectively distinguished between positive and negative samples. The kappa coefficient is 0.9819. This high coefficient indicates the excellent performance of the proposed model. It is worth noting that these exceptional results were obtained by reducing the complexity through the selection of only ten channels, emphasizing the significance of this approach.

To evaluate the accuracy on a per-patient basis, we illustrate the process using the GCN-LSTM model applied to the CHB-MIT-EEG dataset. The accuracy was determined for each file listed in Table 7 by comparing the predicted outputs with the corresponding ground-truth values. These accuracy measurements were recorded for further analysis. Furthermore, to gain deeper insights into the model’s performance, additional metrics, such as recall, precision, F1-measure, and sensitivity, were computed for each patient, as presented in Table 7. The patient IDs are listed in the first column of Table 2. The results are categorized by individual cases identified by their “Case ID”. This table includes information on the number of seizures per case (No. of Seizures) and various performance metrics obtained from testing the model, namely, accuracy, recall, precision, F1-measure, sensitivity, AUC, and kappa. These performance metrics offer a comprehensive overview of a model’s capabilities, allowing researchers to compare different models and select the most effective one for their specific tasks. By providing a detailed breakdown of the model’s performance on a per-patient basis, researchers can assess its suitability and make informed decisions regarding their analysis.

### 3.2. Performance Evaluation on Identifying Seizure State

This series of experiments employed both a learning curve and receiver operating characteristic (ROC) curve to assess the classification performance. In machine learning, the learning curve serves as a sanity check to monitor the training progress of the classifier, whereas the ROC curve, along with its corresponding area under the curve (AUC) value, is utilized to evaluate classification and detection outcomes.

Figure 13 illustrates the learning curve of the classifier during training of the CHB-MIT-EEG dataset. Here, “accuracy” represents the accuracy during the training stage while “valid” indicates the accuracy during the validation stage. This curve is instrumental in verifying the generalization ability of the classifier. Notably, in the training stage, the classifier accuracy on both the training and validation sets remained consistent, with no discernible gap between the curves. Furthermore, the excellent classification performance of the GCN-LSTM model on the test set (accuracy of 99%) indicates that the classifier demonstrated a desirable generalization ability in the current case study, without encountering overfitting or underfitting issues [5].

Figure 14 presents the ROC curve for identifying the state of depression in the MPHC dataset. In situations with an imbalance between positive and negative samples, the ROC curve (AUC value) emerges as a more reliable indicator for evaluating the model’s quality than the precision–recall curve. The high AUC (value = 1) in this context signifies that the proposed classifier effectively distinguished the seizure state.

### 3.3. Effect of Node Embeddings on Model Performance

Figure 15 displays scatter plots depicting the node embeddings acquired by the GCN-LSTM and GCN-BRF models on the CHB-MIT and SSE datasets. To facilitate visualization, the embeddings were reduced to two dimensions using t-SNE, with each point in the plot representing a node from the dataset. The colors assigned to the points reflect the true class labels of the nodes, with each class being represented by a distinct color (yellow for the true class and black for the false class).

The models exhibited remarkable accuracy for both datasets, demonstrating the efficacy of the learned embeddings in capturing the underlying structure of the graph. The scatter plots serve as evidence of the models’ ability to accurately separate and differentiate between different classes, as denoted by the distinguishable clusters formed by the points in different colors. This indicates that the learned embeddings effectively encoded important information about the nodes, allowing the models to capture the intrinsic characteristics of the graph and make accurate predictions.

### 3.4. Confusion Matrix

Figure 16 depicts the confusion matrices of the CHB-MIT, SSE, and TUH databases with the GCN-LSTM and GCN-BRF models. The matrices represent the model’s ability to correctly predict the true positive and true negative classes as well as the false positive and false negative classes.

### 3.5. Comparison of the Models’ Performance with the Literature

Table 8 presents a comprehensive comparison of our models with various models proposed in the literature for seizure detection in EEG signals. The table includes information on the authors, methods employed, and the sensitivity, specificity, and accuracy achieved by each model.

Notably, the proposed GCN-LSTM and GCN-BRF models exhibit superior performance when compared to traditional machine learning models, such as linear discriminant analysis and naive Bayesian, as well as more recent deep learning models, such as 1D-CNN and Bi-GRU network. This suggests that the GCN-LSTM and GCN-BRF models surpass the existing approaches in accurately identifying seizures within EEG signals, as indicated by their higher sensitivity, specificity, and overall accuracy.

## 4. Conclusions

In this study, we propose two methods for detecting seizures in EEG signals, which yielded remarkable results. The approach involved a preprocessing step in which frequency-based, statistical, and Daubechies wavelet transform features were extracted using a 30% overlapping window. These features were then used to create a graphical representation of the EEG signals, which were fed into the GCN model for training and processing. The GCN model captured the complex relationships between the nodes in the graph, resulting in improved seizure detection accuracy. The final step involved using an LSTM classifier, which achieved a 99% accuracy in detecting seizures in the GCN-LSTM model. The GCN-BRF model, which uses a random forest classifier, also achieved 99% accuracy on the CHB-MIT-EEG, SSE-EEG, and TUH-EEG databases. Our results outperformed traditional and recent machine learning models, demonstrating the potential of the GCN-LSTM and GCN-BRF models in accurately detecting seizures in EEG signals. Moreover, we introduce an innovative circular graph visualization method for EEG data that offers a more intuitive and interpretable representation. This novel visualization facilitates effective classification into ictal and interictal classes, achieving heightened accuracy through visual inspection and significantly enhancing the practicality of seizure detection. As a prospective avenue, the integration of visualization in graph clustering and graph oversampling techniques holds promise. The application of advanced graph-based methods for clustering has the potential to refine the grouping of similar patterns within the data, thereby assisting in more nuanced discrimination between seizure and non-seizure signals. Furthermore, the exploration of graph oversampling methods may address imbalances in the dataset, further enhancing the capability of the model to discern subtle distinctions between the two signal types.

The implications of these findings are significant for EEG signal analysis and suggest that graph neural networks, specifically the proposed GCN-LSTM and GCN-BRF models, may be applied to other EEG signal problems beyond seizure detection. By utilizing graph representations and deep learning models, researchers and clinicians can develop more accurate and effective methods for diagnosing and treating various neurological conditions. Moreover, this study demonstrates the potential of graph-based approaches in other fields of research and analysis, particularly in complex systems with interdependent components. Overall, this study presents a promising avenue for future research in the field of EEG signal analysis.

## Figures and Tables

**Figure 1 biomedicines-12-01283-f001:**
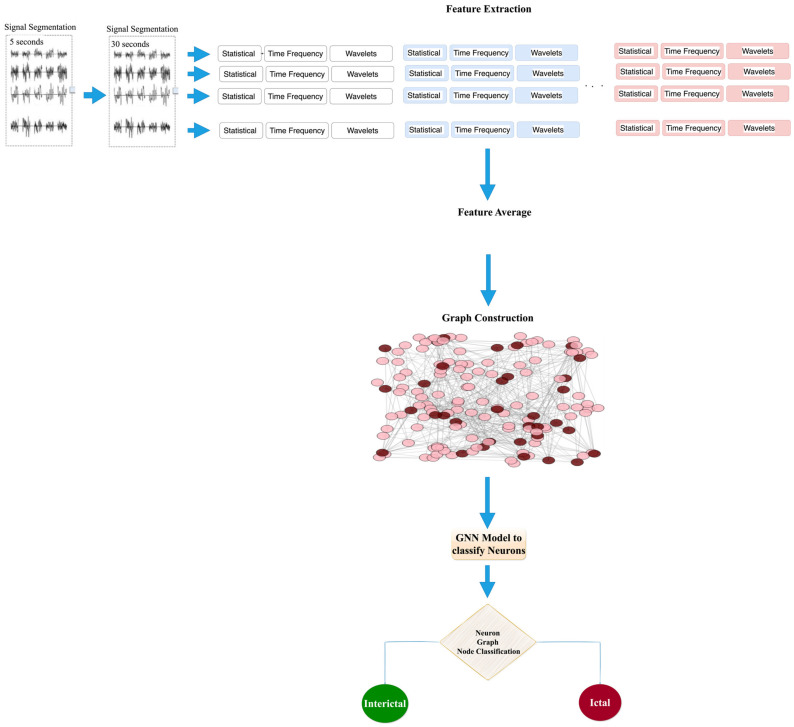
Architectural representation of proposed Seizure detection model using GNN with Seizure EEG signals. Visual presentation of the step-by-step process of the EEG input for classifying epileptic seizures. The GCN-LSTM model combines Graph Convolutional Networks (GCNs) and Long Short-Term Memory (LSTM) networks to effectively handle both graph-based and sequential data. The GCN extracts and enhances features from graph-based data, while LSTM models the temporal dependencies in the data. The model integrates these components into one network to provide a comprehensive representation. Above illustrates the use of the GCN-LSTM model for seizure detection. To improve classification performance, a Balanced Random Forest (BRF) classifier is applied to the output of the GCN model. The GCN model is trained using training data, and the trained model is then used to extract features from test data. These extracted features are subsequently fed into the BRF classifier for classification.

**Figure 2 biomedicines-12-01283-f002:**
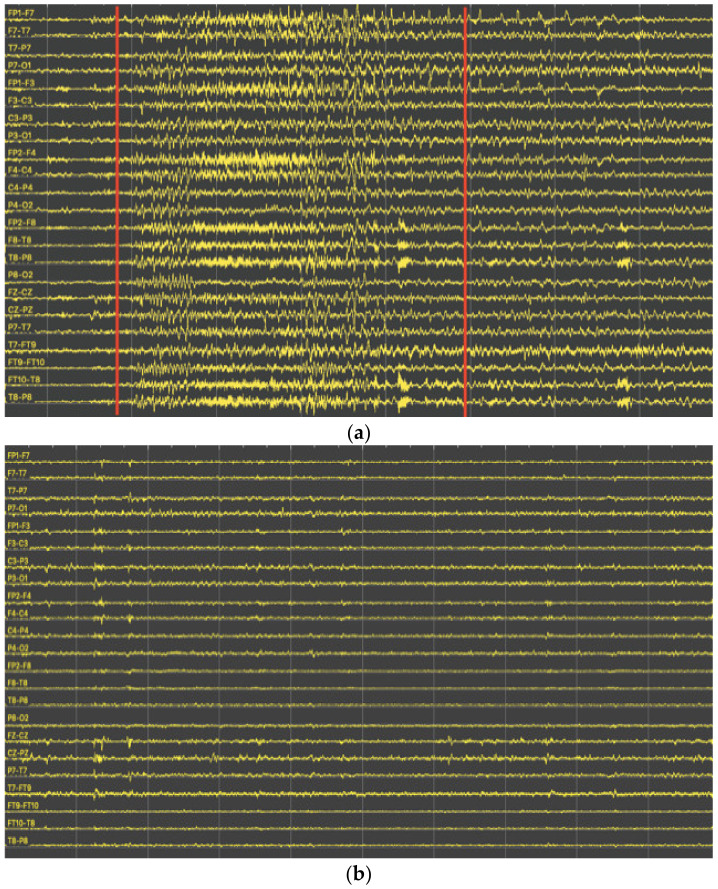
CHB01_03 file with seizure and non-seizure signals. (**a**) shows abnormal electrical activity in the brain, consistent with a seizure. The figure shows a spike and wave pattern, which is a characteristic of a seizure. The spike and wave patterns are caused by abnormal synchronization of electrical activity in the brain. This synchronization can lead to a seizure, which is a sudden burst of electrical activity in the brain that can cause a variety of symptoms, including loss of consciousness, convulsions, and sensory disturbances. The red lines correspond to the windows used. (**b**) shows normal electrical activity in the brain. The figure does not show any spike and wave patterns, which is consistent with normal brain activity. Normal brain activity is characterized by a variety of different electrical patterns, including alpha waves, beta waves, theta waves, and delta waves. These waves are associated with different levels of consciousness and brain activity. For example, alpha waves are associated with a relaxed state of wakefulness, while beta waves are associated with a state of alertness.

**Figure 3 biomedicines-12-01283-f003:**
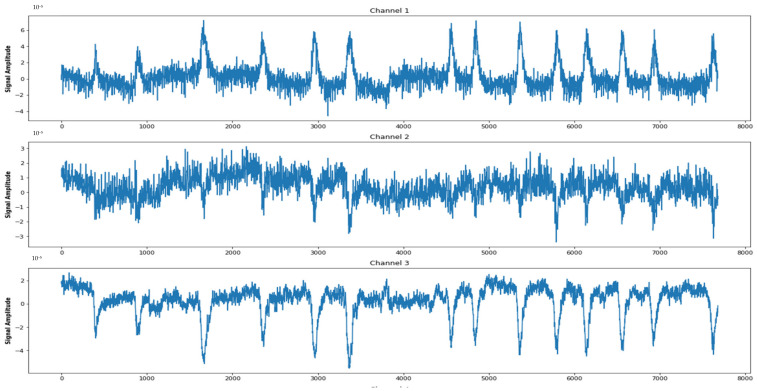
First three channels from PN00-1 file. We have displayed the EEG file PN00-1 from the Siena Scalp EEG Database. This particular file consists of 34 channels, capturing electrical activity from various regions of the scalp. For visualization purposes, we have chosen to display the first three channels of the EEG file.

**Figure 4 biomedicines-12-01283-f004:**
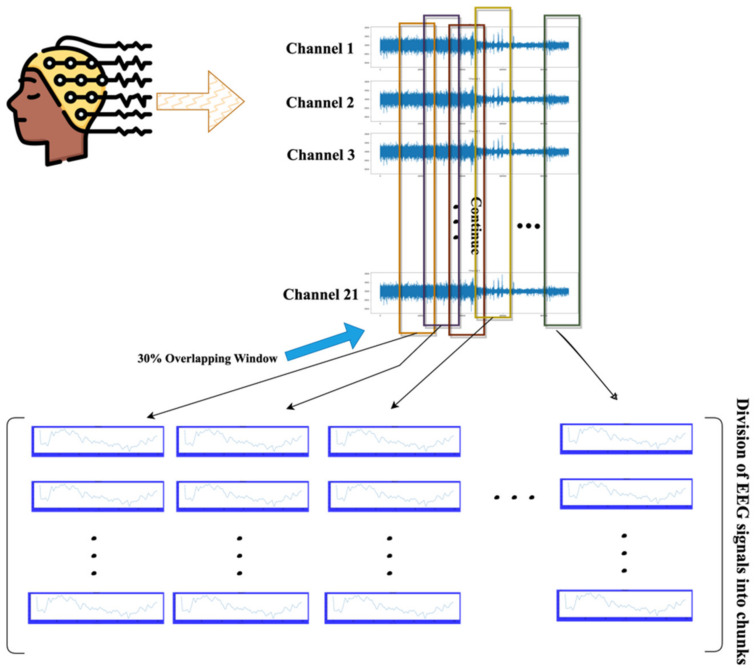
Division of EEG signals into sequences of overlapping windows of 30 s with a 30% overlapping window. The use of a 30 s window with a 30% overlap on multichannel EEG signals has been chosen to strike a balance between preserving temporal information and minimizing the impact of windowing artifacts. The 30% overlapping ratio ensures that the signal chunks retain continuity, thereby reducing the likelihood of information loss at the edges of the windows.

**Figure 5 biomedicines-12-01283-f005:**
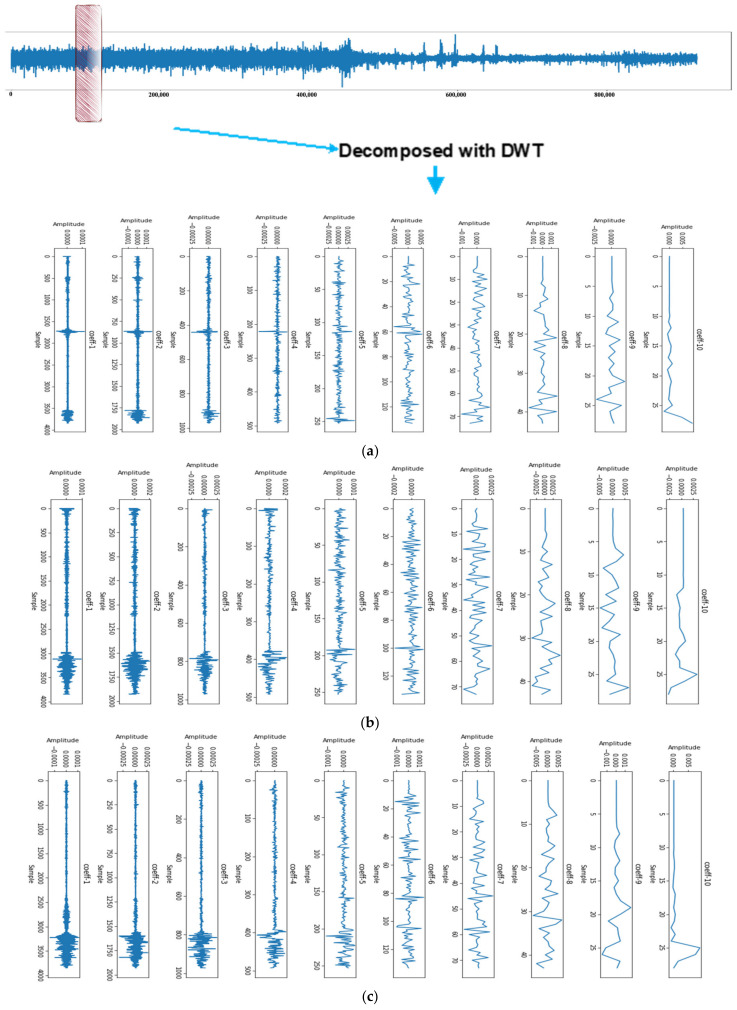
Coefficient Visualization with non-seizure Signal. (**a**) CHB-MIT-EEG. (**b**) SSE-EEG. (**c**) TUH-EEG. We conducted an analysis on three datasets using the Daubechies 8 db wavelet. The figure shows the results for the non-seizure windows. For each frequency subband obtained through the Daubechies wavelet, we created separate subplots arranged in descending order of frequency content. Each subplot is labeled with the number of coefficients being plotted, while the x-axis is labeled as “Sample” and the y-axis as “Amplitude”.

**Figure 6 biomedicines-12-01283-f006:**
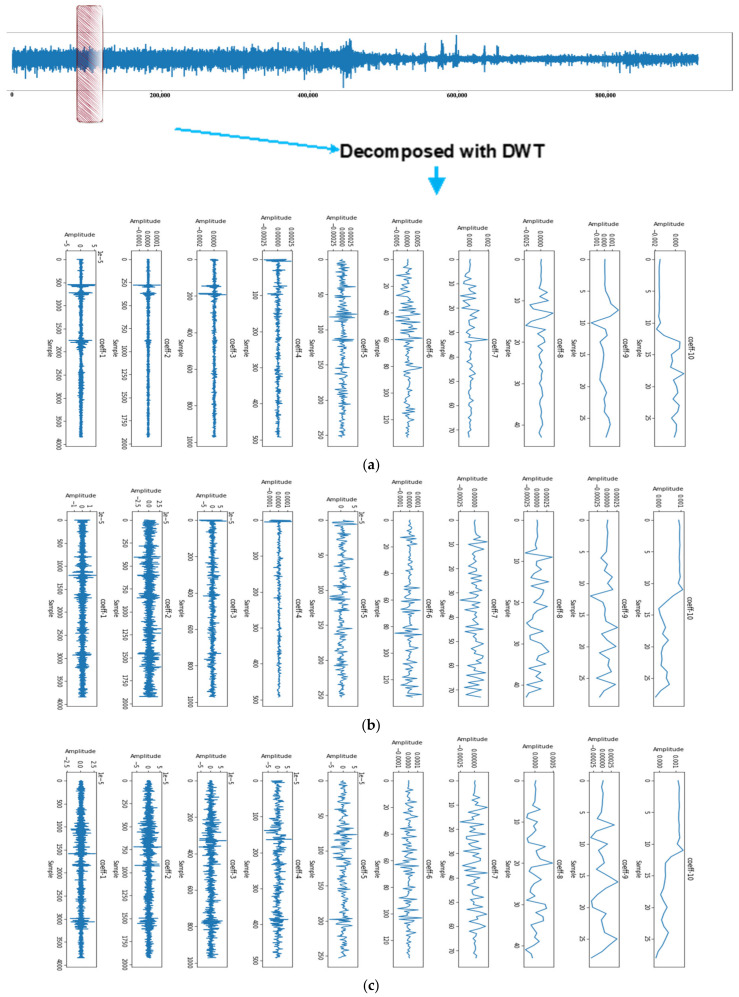
Coefficient Visualization with Seizure Signal. (**a**) CHB-MIT-EEG. (**b**) SSE-EEG. (**c**) TUH-EEG. We conducted an analysis on three datasets using the Daubechies wavelet 8 db. The figure shows the results for the seizure windows. For each frequency subband obtained through the Daubechies wavelet, we created separate subplots arranged in descending order of frequency content. Each subplot is labeled with the number of coefficients being plotted, while the x-axis is labeled as “Sample” and the y-axis as “Amplitude”.

**Figure 7 biomedicines-12-01283-f007:**
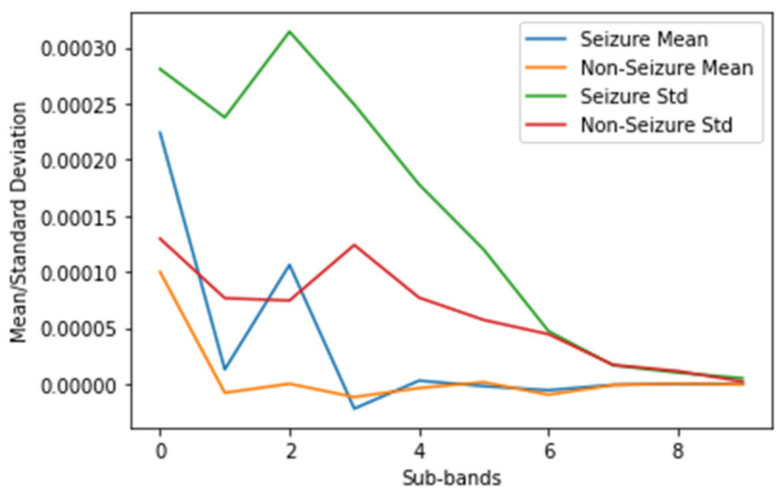
Distinction of Seizure and Non-Seizure EEG Signals through the Mean and Standard Deviation of Energy from subbands using the Daubechies Wavelet Transform. The EEG signals were subjected to the wavelet transform to capture their distinctive features. The wavelet coefficients were analyzed in two separate windows, each representing seizure and non-seizure signals, to determine their mean and standard deviation.

**Figure 8 biomedicines-12-01283-f008:**
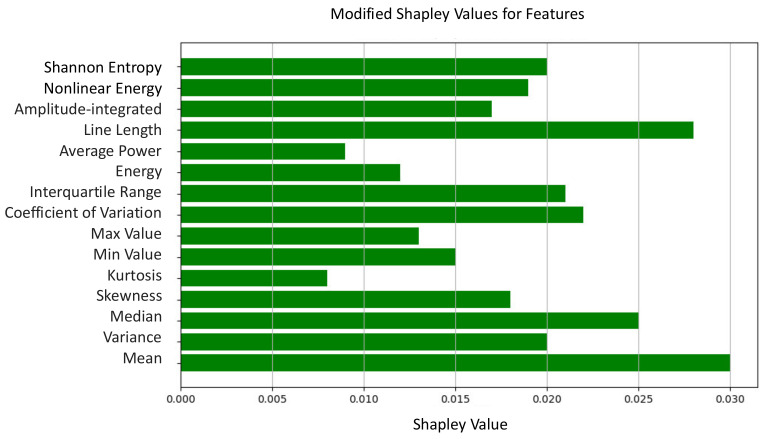
Shapley value. Presentation of the modified Shapley values for the different features used.

**Figure 9 biomedicines-12-01283-f009:**
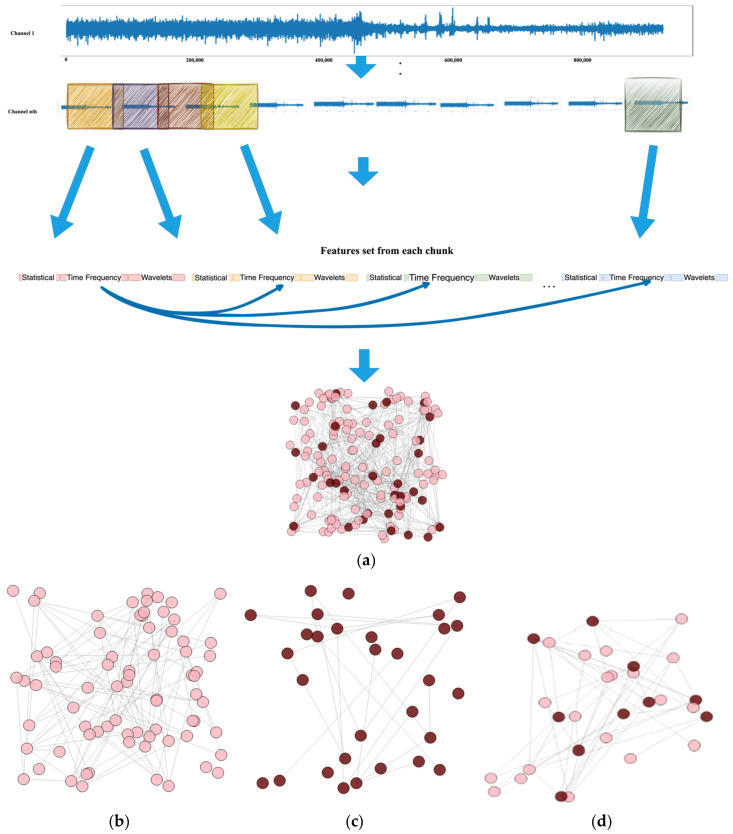
Graph representation. (**a**) Process of converting Feature set into Graph representation. (**b**) Graph node classification for non-seizure signal (light-colored circle). (**c**) Graph node classification for seizure signal (dark-colored circle). (**d**) Graph node classification for Seizure and non-seizure samples.

**Figure 10 biomedicines-12-01283-f010:**
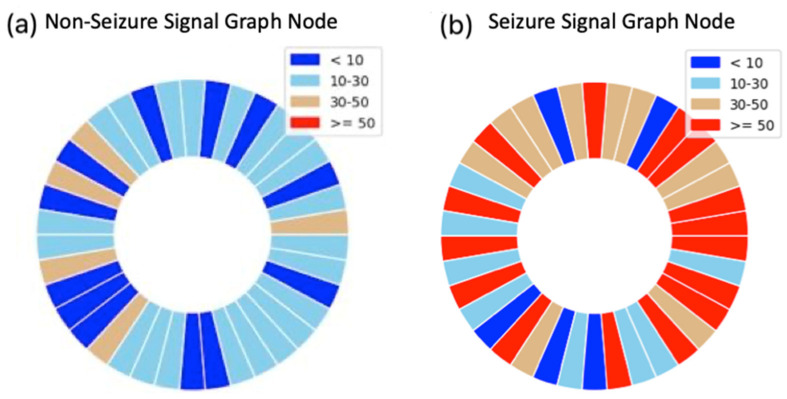
Graph representation. (**a**) Non-seizure signal node feature set with 38 selected Features. (**b**) Neurological disease signal node feature set with 38 selected Features. (**a**,**b**) illustrate the features extracted from non-seizure and seizure signals, respectively. The seizure node values are slightly higher than those of the non-seizure signal, as shown in the plot.

**Figure 11 biomedicines-12-01283-f011:**
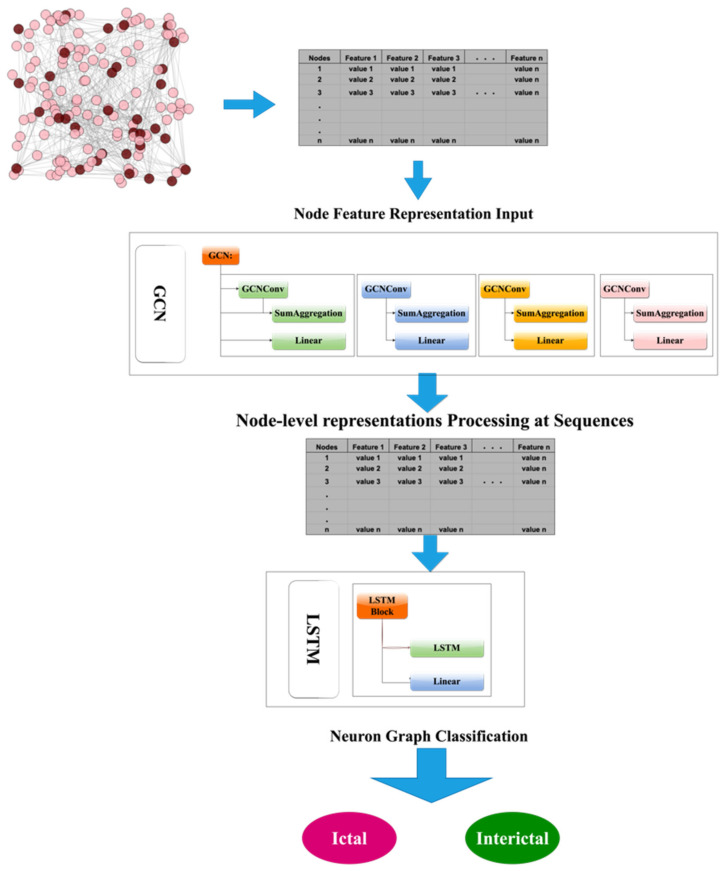
GCN-LSTM model diagram. The GCN component uses node features and edge connectivity to create node-level representations of the graph’s structure. It applies non-linear activation functions through layers to capture both local and global information. The LSTM component takes these node-level representations and processes them over time, using gates to regulate information flow and remembering past representations. The final hidden state summarizes the sequence. A linear layer transforms the hidden state to generate the final prediction based on the graph data.

**Figure 12 biomedicines-12-01283-f012:**
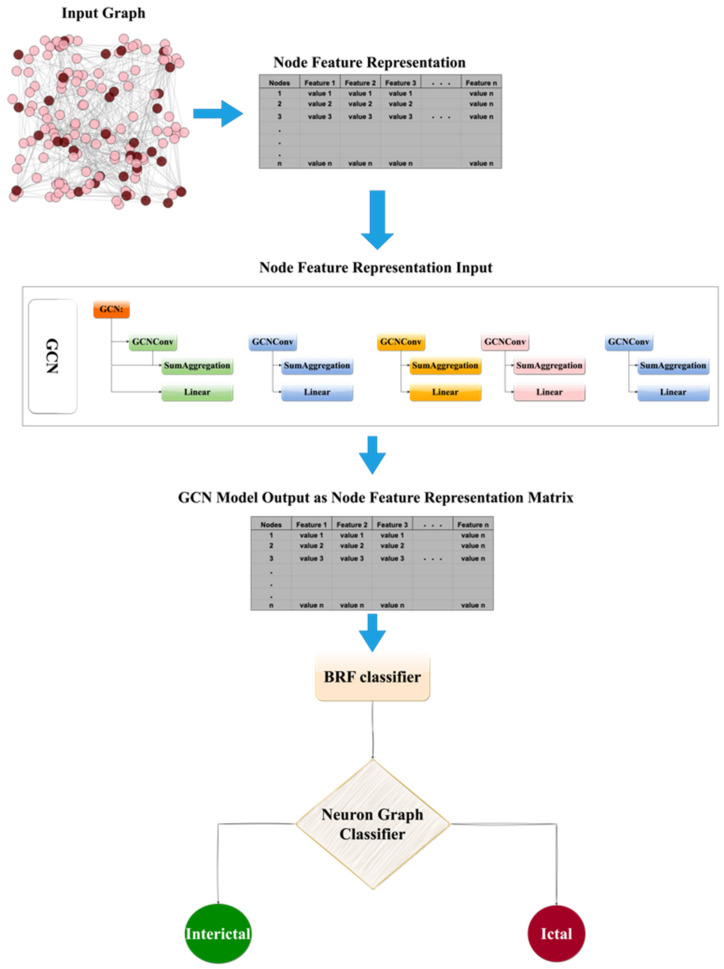
GCN-BRF model diagram. The GCN-BRF model combines the strengths of GCN for learning graph representations and BRF for classification, resulting in enhanced performance on graph classification tasks. The model initially employs the GCN operation to capture temporal dependencies within the output features. Ultimately, the BRF classifier is utilized to generate the final prediction.

**Figure 13 biomedicines-12-01283-f013:**
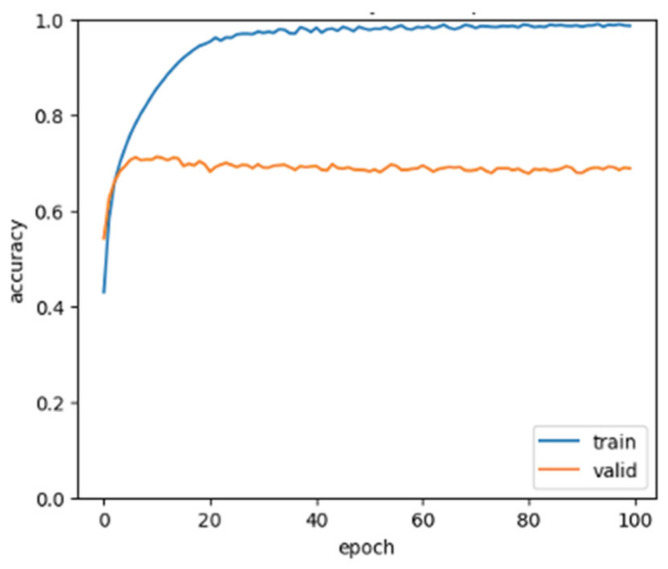
Learning curve on identifying seizure state. Performance of GCN-LSTM model over 3000 windows, showing the accuracy of the binary classification model on the training and validation datasets. The graph indicates that the model achieves a high accuracy of 99%, demonstrating its ability to classify data with precision.

**Figure 14 biomedicines-12-01283-f014:**
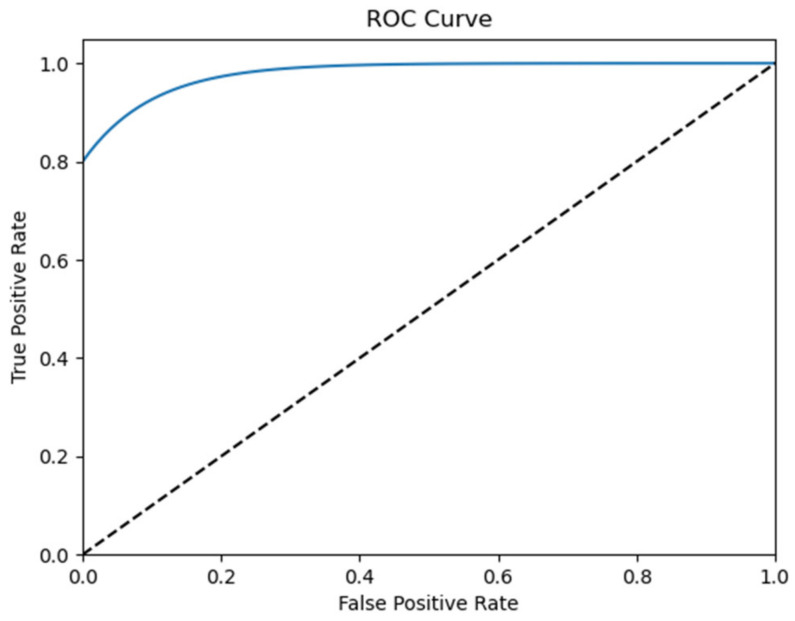
ROC curve. The ROC curve on identifying seizure state from CHB-MIT Dataset.

**Figure 15 biomedicines-12-01283-f015:**
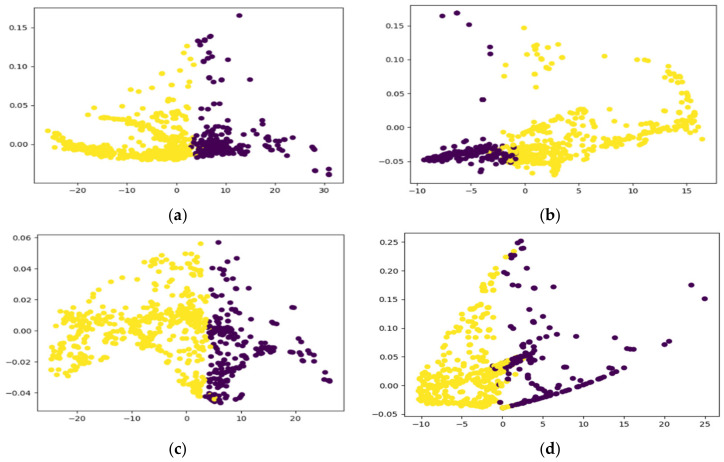
Final model performance. (**a**,**b**) demonstrate the node embeddings produced by the GCN-LSTM model on the CHB-MIT and SSE datasets, respectively. (**c**,**d**) depict the node embeddings generated by the GCN-BRF model on the CHB-MIT and SSE datasets, respectively. The embeddings were reduced to two dimensions using t-SNE, and each point represents a node in the dataset.

**Figure 16 biomedicines-12-01283-f016:**
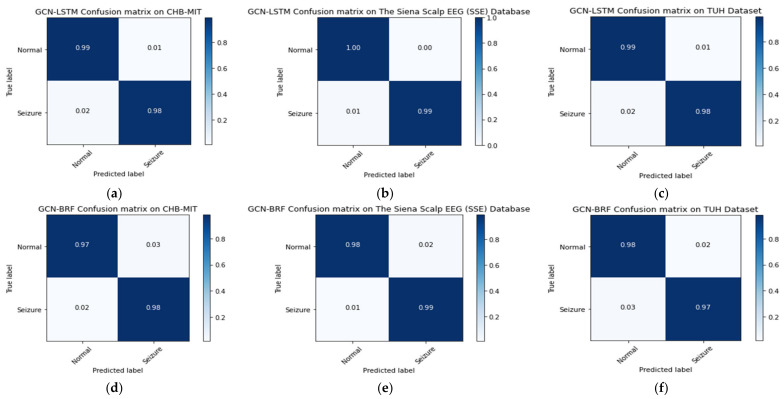
Confusion matrix. (**a**–**c**) Evaluation of performance of the GCN-LSTM model on CHB-MIT-EEG, SSE-EEG, and TUH-EEG databases using confusion matrix. (**d**–**f**) Evaluation of performance of the GCN-BRF model on CHB-MIT-EEG, SSE-EEG, and TUH-EEG databases using confusion matrix.

**Table 1 biomedicines-12-01283-t001:** Research Statement Summary.

Study	Method	Datasets	Results	Challenges
Gómez et al. [16]	FCN	CHB-MIT-EEG	Attaining average accuracy and specificity rates of 99.3% and 99.6%, respectively, for the CHB-MIT dataset, as well as corresponding rates of 98.0% and 98.3% for the EPILEPSIAE patients.	Exploring an automated approach for epileptic seizure detection by employing an imaged-EEG representation of brain signals.
Dissanayake et al. [38]	Deep learning	CHB-MIT-EEG	The proposed models achieve a state-of-the-art level of performance for seizure prediction on the CHB-MIT-EEG dataset, exhibiting accuracies of 88.81% and 91.54%, respectively.	Designing patient-independent seizure prediction models that can adapt to the considerable inter-subject variability present in EEG data.
Zhang et al. [39]	CSP and CNN	CHB-MIT-EEG	The model attains an accuracy rate of 92.2%.	Distinguishing between the preictal and interictal states, as their high similarity complicates seizure prediction, unlike the relatively well-studied seizure detection that targets the interictal and ictal states.
Tsiouris et al. [26]	RIPPER, SVM, NN	CHB-MIT-EEG	The SVM classifier demonstrates the highest classification accuracy in the patient-specific scenario, achieving 85.75% sensitivity and specificity. However, in the patient-independent case, its performance is comparatively lower.	Exploring the capability of various classification methods to differentiate between preictal and interictal EEG segments.
Chen et al. [40]	Autoregressive moving average and one-class SVM	Bern-Barcelona and CHB-MIT EEG	Introducing a unified framework for early seizure detection and epilepsy diagnosis, which achieves classification accuracies of 93% and 94%.	Presumptions concerning the stationarity, linearity, and normality of EEG time series.
Zabihi et al. [27]	Phase-space representation and LDA and naive Bayesian classifiers	CHB-MIT benchmark database	Obtaining an average sensitivity of 88.27% and specificity of 93.21% in the detection of epileptic seizures.	Examining how well the phase-space representation can elucidate the fundamental dynamics of epileptic seizures.
Zhou et al. [41]	CNN	Intracranial Freiburg and scalp CHB-MIT	Frequency-domain signals outperformed time-domain signals in detecting epileptic seizures, with an average accuracy of 96.7–95.6% in Freiburg and 95.4–97.5% in CHB-MIT.	Accurately characterizing signal properties and effectively distinguishing between ictal, preictal, and interictal segments.
Shoka et al. [42]	CNN	CHB-MIT-EEG	Achieving accuracy levels of up to 86.11% and 84.72% when employing GoogLeNet in combination with Arnold and chaotic methods.	Providing a robust approach for encrypted EEG classification and prediction.
Chou et al. [43]	CNN	CHB-MIT-EEG	The peak performance was observed during testing with only 1% of the testing dataset, resulting in an overall accuracy of 85.9%. The accuracy for the ictal stage was notably high at 97.7%.	Leveraging state-of-the-art deep learning techniques to improve the efficiency of critical stage detection in EEG for epilepsy diagnosis.
Jácobo-Zavaleta and Zavaleta [44]	ChronoNet	TUH-EEG	Achieving sensitivities of up to 71.50% and specificities of up to 83.70% for patient-control detection, while reaching sensitivities of up to 56.60% and specificities of up to 95.90% for patient-specific detection.	Identifying the optimal performance for seizure detection using raw EEG signals from the TUH EEG Seizure Corpus database.
Ahmedt-Aristizabal et al. [45]	Deep learning model	TUH-EEG	In a multifold cross-validation of the region-based approach, an average test accuracy of 95.19% and an average AUC of 0.98 for the ROC curve were observed. In contrast, when applying a leave-one-subject-out cross-validation scheme for the same approach, accuracy declined due to data limitations, resulting in an average test accuracy of 50.85%.	Seeking to automatically extract and categorize semiological patterns from facial expressions, addressing the limitations of current computer-based analytical methods in epilepsy monitoring that have predominantly overlooked facial movements.
Dissanayake et al. [46]	Geometric deep learning	SSE	The models introduced in both stages attain a state-of-the-art performance by using a one-hour early seizure prediction window on two benchmark datasets. Specifically, they achieve an accuracy of 95.38% with 23 subjects on the CHB-MIT-EEG dataset and 96.05% with 15 subjects on the Siena-EEG dataset.	Exploring seizure prediction in scenarios where the target subject has limited or no training data available.
Sánchez-Hernández et al. [47]	Feature selection methods	SSE	The highest F1-score of 90% was achieved by the K-nearest neighbor algorithm in conjunction with the CHB-MIT dataset.	Employment of feature selection techniques to choose features that enhance pattern recognition in the detection of ictal activity from the CHB-MIT and Siena Scalp EEG databases.
Fatlawi and Kiss [48]	SAW	SSE	The model attains an accuracy rate of 96.44%.	Tackling the issue of unbalanced representation among classification targets in data streams.

**Table 2 biomedicines-12-01283-t002:** Children’s Hospital Boston-Massachusetts Institute of Technology Electroencephalogram (CHB-MIT-EEG).

Patient	Gender	Age	Number of Seizure Events (Tmax–Tmin in seconds)	Total Duration of Seizures (s)	Total Duration of Non-Seizures (s)	Total Duration (s)
1	F	11	7 (28–102)	449	23,476	23,925
2	M	11	3 (10–83)	175	7984	8159
3	F	14	7 (48–70)	409	24,791	25,200
4	M	22	4 (50–117)	382	37,977	38,359
5	F	7	5 (97–121)	563	17,437	18,000
6	F	1.5	10 (13–21)	163	93,053	93,216
7	F	14.5	3 (87–144)	328	32,209	32,537
8	M	3.5	5 (135–265)	924	17,076	18,000
9	F	10	4 (63–80)	280	34,219	34,499
10	M	3	7 (36–90)	454	50,010	50,464
11	F	12	3 (23–753)	809	9250	10,059
12	F	2	27 (14–98)	1016	33,844	34,860
13	F	3	10 (18–71)	450	24,750	25,200
14	F	9	8 (15–42)	177	25,023	25,200
16	F	7	8 (7–15)	77	17,923	18,000
17	F	12	3 (89–116)	296	10,528	10,824
18	F	18	6 (31–69)	323	19,951	20,274
19	F	19	3 (78–82)	239	10,307	10,546
20	F	6	8 (30–50)	302	19,734	20,036
21	F	13	4 (13–82)	203	13,587	13,790
22	F	9	3 (59–75)	207	10,593	10,800
23	F	6	7 (21–114)	431	31,823	32,254
24	Unknown	Unknown	16 (17–71)	539	42,661	43,200

**Table 3 biomedicines-12-01283-t003:** Description of EEG Datasets: Number of Subjects, Gender Distribution, Age Range, Seizure Events, Seizure Duration, EEG Channels, and Sampling Rate.

Dataset	Number of Subjects	Gender Distribution	Age Range (Years)	Seizure Events	Seizure Duration	EEG Channels	Sampling Rate (Hz)
CHB-MIT-EEG	22	10 male, 12 female	0.3–17.5	198	A few seconds to several minutes	22	256
SSE-EEG	14	9 male, 5 female	20–71	47	A few seconds to few minutes	Varies (up to 34)	512
TUH-EEG	10,874	51% female	<1–90+	N/A (ongoing collection)	N/A	Varies (up to 31)	Most at 250

**Table 4 biomedicines-12-01283-t004:** GCN-LSTM Summary.

Layer (Type)	Output Shape	Amount of Params
GCN		
conv1 (GCNConv)	(N, 256, 1400)	131,072
ReLU	(N, 256, 1400)	0
conv2 (GCNConv)	(N, 256, 256)	524,544
ReLU	(N, 256, 256)	0
conv3 (GCNConv)	(N, 256, 256)	524,544
ReLU	(N, 256, 256)	0
conv4 (GCNConv)	(N, 256, 256)	524,544
ReLU	(N, 256, 256)	0
conv5 (GCNConv)	(N, 16, 256)	40,976
ReLU	(N, 16, 256)	0
LSTM		
lstm (LSTM)	(1, 256, 16)	330,240
fc (Linear)	(1, 2)	514
Total params: 2,077,434 Trainable params: 2,077,434 Non-trainable params: 0		

**Table 5 biomedicines-12-01283-t005:** GCN-BRF Summary.

Layer (Type)	Output Shape	Amount of Params
conv1 (GCNConv)	(N, 256, 1400)	131,072
ReLU	(N, 256, 1400)	0
conv2 (GCNConv)	(N, 256, 256)	524,544
ReLU	(N, 256, 256)	0
conv3 (GCNConv)	(N, 256, 256)	524,544
ReLU	(N, 256, 256)	0
conv4 (GCNConv)	(N, 256, 256)	524,544
ReLU	(N, 256, 256)	0
conv5 (GCNConv)	(N, 16, 256)	40,976
ReLU	(N, 16, 256)	0
Total params: 1,746,680 Trainable params: 1,746,680 Non-trainable params: 0		

**Table 6 biomedicines-12-01283-t006:** Results of Accuracy, Recall, Precision, F1-measure, Sensitivity, AUC, and Kappa for the GCN-LSTM and GCN-BRF models on CHB-MIT-EEG, SSE, and TUH datasets.

GCN-LSTM Model									
Datasets	CHB-MIT-EEG			SSE-EEG			TUH-EEG		
Metrics	Without Augmentation	SMOTE	KNNOR	Without Augmentation	SMOTE	KNNOR	Without Augmentation	SMOTE	KNNOR
Accuracy	0.9700	0.9834	**0.9973**	0.9746	0.9941	**0.9985**	0.9652	0.9748	**0.9958**
Recall	0.9435	0.9679	**0.9823**	0.9552	0.9810	**0.9825**	0.9685	0.9858	**0.9947**
Precision	**1.0**	**1.0**	0.9874	0.9946	0.9862	**1.0**	0.9752	**0.9825**	**0.9858**
F1-measure	0.9709	0.9837	**0.9874**	0.9745	**0.9868**	0.9855	0.9625	**0.9858**	0.9755
Sensitivity	0.9596	0.9650	**0.9865**	0.9552	**0.9874**	0.9800	0.9525	**0.9758**	0.9747
AUC	0.9719	0.9858	**0.9911**	0.9749	0.9900	**0.9952**	0.9858	0.9858	**0.9925**
Kappa	0.9437	0.9600	**0.9879**	0.9492	**0.9863**	0.9840	0.9745	**0.9698**	**0.9698**
**GCN-BRF Model**									
**Datasets**	**CHB-MIT-EEG**			**SSE-EEG**			**TUH-EEG**		
**Metrics**	**Without Augmentation**	**SMOTE**	**KNNOR**	**Without Augmentation**	**SMOTE**	**KNNOR**	**Without Augmentation**	**SMOTE**	**KNNOR**
Accuracy	0.9845	0.9928	**0.9961**	0.9735	0.9900	**0.9952**	0.9702	0.9711	**0.9921**
Recall	0.9851	0.9795	**0.9800**	0.94965	0.9825	**0.9852**	0.9517	0.9885	**0.9985**
Precision	0.9852	**0.9890**	0.9790	0.9801	0.9802	**0.9985**	0.9785	0.9839	**0.9893**
F1-measure	0.9862	0.9798	**0.9885**	0.9689	0.9798	**0.989**	0.9689	**0.9782**	0.9710
Sensitivity	0.98	0.9851	**0.9885**	0.9601	**0.9874**	0.9700	0.9560	0.9663	**0.9734**
AUC	0.98	0.9898	**0.9902**	0.9752	**0.9895**	0.9850	0.96006	0.9754	**0.9817**
Kappa	0.9801	0.9885	**0.9896**	0.9562	**0.9960**	**0.9690**	0.9658	**0.9655**	0.9622

**Table 7 biomedicines-12-01283-t007:** Results of Accuracy, Recall, Precision, F1-measure, Sensitivity, AUC, and Kappa for the GCN-LSTM model on CHB-MIT-EEG dataset.

Case ID	No. of Seizures	Accuracy	Recall	Precision	F1-Measure	Sensitivity	AUC	Kappa
CHB1	7	0.9714	0.9429	1	0.9706	0.9429	0.9715	0.9429
CHB2	3	0.9708	0.9429	0.9986	0.97	0.9419	0.9708	0.9416
CHB3	7	0.9708	0.9429	0.9986	0.97	0.9429	0.9416	0.9708
CHB4	4	0.9695	0.9429	0.9959	0.9687	0.9429	0.939	0.9695
CHB5	5	0.9695	0.939	1	0.9686	0.939	0.939	0.9695
CHB6	10	0.9701	0.9403	1	0.9693	0.9403	0.9403	0.9702
CHB7	3	0.9682	0.9364	1	0.9672	0.9364	0.9364	0.9682
CHB8	5	0.9552	0.9364	0.973	0.9544	0.9364	0.9104	0.9552
CHB9	4	0.9669	0.9429	0.9905	0.9661	0.9429	0.9338	0.9669
CHB10	7	0.9718	0.9429	0.9986	60.97	0.9429	0.9416	0.9708
CHB11	2 (3)	0.9656	0.9377	0.9931	0.9646	0.9377	0.9312	0.9656
CHB12	21 (40)	0.9623	0.9377	0.9864	0.9614	0.9377	0.9247	0.9624
CHB13	10 (12)	0.9701	0.9403	1	0.9693	0.9403	0.9403	0.9702
CHB14	8	0.9508	0.9257	0.9747	0.9496	0.9257	0.9017	0.9508
CHB15	17 (20)	0.95	0.9222	0.9766	0.9486	0.9222	0.9	0.95
CHB16	9 (10)	0.9494	0.9196	0.9779	0.9479	0.9196	0.8987	0.9494
CHB17	3	0.9692	0.9429	0.9952	0.9684	0.9429	0.92405	0.98345
CHB18	6	0.96863	0.9429	0.99397	0.96783	0.943	0.91138	0.99435
CHB19	3	0.96806	0.9429	0.99274	0.96726	0.9431	0.89871	0.998
CHB20	8	0.96749	0.9429	0.99151	0.96669	0.9432	0.88604	1.000
CHB21	4	0.96692	0.9429	0.99028	0.96612	0.9433	0.87337	0.9960
CHB22	3	0.96635	0.9429	0.98905	0.96555	0.9434	0.8607	0.9858
CHB23	7	0.96578	0.9429	0.98782	0.96498	0.9435	0.84803	0.9957

**Table 8 biomedicines-12-01283-t008:** Comparison of Model Performance: A Comprehensive Study of Previous Approaches and Our Proposed GCN-LSTM and GCN + BRF Models.

Dataset	Author	Method	Sensitivity	Specificity	Accuracy
CHB-MIT-EEG	Kiranyaz et al. [63]	Collective network of binary classifiers	89.01	94.71	–
	Zabihi et al. [27]	Linear discriminate analysis + Naive Bayesian	89.1	94.8	94.69
	Samiee et al. [64]	Sparse rational decomposition + Local Gabor binary patterns	70.4	99.1	83.53
	Liang et al. [65]	CNN + LSTM	84	99	99
	Hu et al. [66]	Local mean decomposition + Bi-LSTM	93.61	91.85	-
	Tsiouris [67]	EEG features + LSTM network	99.38	99.6	-
	Yang et al. [68]	STFT spectral images + Dual self-attention residual network	89.25	92.67	92.07
	Wang et al. [69]	Stacked 1D-CNN	88.14	99.62	99.54
	Peng et al. [70]	Stein-kernel based sparse representation	97.85	98.57	98.21
	Shoka et al. [71]	Channel selection + Ensemble classifier	100	77.5	89.02
	Zhang [72]	Bi-GRU network	93.89	98.49	98.49
	Proposed method	GCN + BRF	98.86	98.00	99.61
	Proposed method	GCN + LSTM	98.65	98.00	99.73
SEE-EEG	Dissanayake [46]	Geometric deep learning	95.88	96.41	95.88
	Sergio E et al. [47]	Feature selection methods	76	-	96
	Attila Kiss [48]	SAW	93.13	96.66	96.44
	Proposed method	GCN + BRF	97.00	99.00	99.52
	Proposed method	GCN + LSTM	98.00	97.00	99.85
TUH-EEG	Ahmedt Aristizabal et al. [45]	Sigmoid, 2D CNN-LSTM	71.50	83.70	92.50
	Jorge Zavaleta et al. [44]	LSTM, ChronoNet	56.60	95.90	-
	Proposed method	GCN + BRF	98.69	98.14	99.01
	Proposed method	GCN + LSTM	98.10	97.25	99.40

## Data Availability

The datasets are publicly available from the links https://physionet.org/content/chbmit/1.0.0/, accessed on 29 April 2023, https://physionet.org/content/siena-scalp-eeg/1.0.0/, accessed on 13 May 2023, and https://isip.piconepress.com/projects/tuh_eeg/, accessed on 15 May 2023.

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
