# Peer review of "Graphical Insight: Revolutionizing Seizure Detection with EEG Representation"

_biomedicines, 2024, doi:10.3390/biomedicines12061283_

Round 1

Reviewer 1 Report

Comments and Suggestions for Authors

In the provided manuscript, the authors propose a novel approach to detecting and characterizing epileptic activity in multichannel EEG recordings. Specifically, the authors introduce a combination of Graph Convolutional Networks (GCNs), Long Short-Term Memory (LSTM), and Balanced Random Forest (BRF) for seizure detection. The study employs a feature extraction method that includes frequency-based, statistical-based, and Daubechies wavelet transform features.

The manuscript provides a thorough evaluation of the models using a variety of metrics such as accuracy, sensitivity, and specificity. The inclusion of two different models and their comparison with previous methods adds depth to the analysis.

However, the manuscript presents results from the application of the models but offers limited discussion on the generalizability of the approach to varied neurological conditions. The manuscript also does not sufficiently address the computational demands or the practical implementation challenges of the proposed models, which could be significant given the complexity of GNNs. Moreover, the study primarily focuses on retrospective data from EEG datasets. There is a notable lack of prospective studies or real-world testing, which are crucial for validating the efficacy of the models in clinical settings.

The manuscript would benefit from a more critical discussion of the limitations and potential pitfalls of the proposed models, including scenarios where they might fail or provide misleading outputs. I would strongly suggest updating the text to incorporate the above-mentioned comments.

Minor comments:

- More detailed technical specifications and parameter settings for the GNN models could be provided to ensure reproducibility of the results.

- Additional details on the preprocessing steps applied to the EEG data before feature extraction would clarify the initial data handling process.

- A discussion on how the models handle noise and artifacts in EEG data, which are common issues, would enhance the manuscript's comprehensiveness.

Author Response

Dear Reviewer,

Please find the answers to your comments. We have carefully addressed all the issues you pointed out and made corresponding modifications to the manuscript. The updated text is highlighted in yellow.

We would like to thank you for your constructive and relevant comments, as well as for your suggestions.

Best regards.

The authors.

Reviewer 2 Report

Comments and Suggestions for Authors

This paper developed two models to enhance seizure detection accuracy in EEG signals utilizing GCN and LSTM, GCN and BRF. However, the current version of the manuscript is premature and needs further effort to be potentially suitable for publication in biomedicine.  

1. The paper cited a large number of literature reviews and corresponding table analyses, but it does not emphasize the advantages of the models used in this study or the research gaps.  

2.Fig. 2 does not clearly show the characteristic EEG signals during seizures. Please provide an enlarged image to better demonstrate this.  

3. The article mentions the use of SMOTE and KNNOR methods. It is recommended to further explain how these methods are specifically applied to EEG data.

Author Response

(The authors gave the same response as above.)
